# FOXO1 links KRAS G12D and G12V alleles to glutamine and nitrogen metabolism in colorectal cancer

Suzan Ber [ID] [1,2], Ming Yang[1], Marco Sciacovelli[1,3], Shamith Samarajiwa[1], Khushali Patel[1], Efterpi Nikitopoulou[1], Annie Howitt[1], Simon J Cook[2], Ashok R Venkitaraman [ID] [1,4,5,6,7,8], Christian Frezza [ID] [1,9,10] & Alessandro Esposito [ID] [1,11 ✉]

## Abstract

Mutations in KRAS, particularly at codon 12, are frequent in adenocarcinomas of the colon, lungs and pancreas, driving carcinogenesis by altering cell signalling and reprogramming metabolism. However, the specific mechanisms by which different KRAS G12 alleles initiate distinctive patterns of metabolic reprogramming are unclear. Using isogenic panels of colorectal cell lines harbouring the G12A, G12C, G12D and G12V heterozygous mutations and employing transcriptomics, metabolomics, and extensive biochemical validation, we characterise distinctive features of each allele. We demonstrate that cells harbouring the common G12D and G12V oncogenic mutations significantly alter glutamine metabolism and nitrogen recycling through FOXO1-mediated regulation compared to parental lines. Moreover, with a combination of small molecule inhibitors targeting glutamine and glutamate metabolism, we also identify a common vulnerability that eliminates mutant cells selectively. These results highlight a previously unreported mutant-specific effect of KRAS alleles on metabolism and signalling that could be potentially harnessed for cancer therapy.

**Keywords** FOXO Signalling; KRAS Mutation; Glutamine Metabolism; Colorectal Cancer; Glutamine Synthase
**Subject Categories** Cancer; Metabolism; Signal Transduction

## Introduction

The three RAS genes, HRAS, NRAS, and KRAS, code for membrane-bound small GTPases that play fundamental roles in development, adult tissue homeostasis, and disease. Mutations in RAS genes and deregulation of RAS-dependent signalling pathways drive several cancers associated with poor prognosis (Ostrow et al, 2016). Missense mutations in RAS genes occur in 25–30% of all cancers, particularly at three hotspots, glycine-12 (G12), glycine-13 (G13), and glutamine-61 (Q61), with a tissue-dependent association. For instance, HRAS mutations occur more often in bladder cancer, NRAS in melanoma and KRAS in colon, lung and pancreatic adenocarcinomas (Haigis, 2017; Cox et al, 2014). The frequency of specific missense mutations within a hotspot also depends on the site of occurrence. For example, the missense KRAS mutations G12D and G12V are frequent in all KRAS-driven cancers, but G12R and G12C substitutions are very prevalent only in pancreatic (~15-20%) and lung (~60%) adenocarcinomas, respectively (Cox et al, 2014).

The analysis of mutational signatures in patients suggests that the selection of mutation-dependent oncogenic signals might trigger distinct phenotypes in permissive tissues (Haigis et al, 2019; Li et al, 2018a; Ostrow et al, 2016). The mechanisms for this selection are largely undetermined, but different mutations at glycine-12 result in altered GTP hydrolysis mediated by KRAS, differential engagement of effector proteins and signalling (Hunter et al, 2015a; Ihle et al, 2012; Munoz-Maldonado et al, 2019; Yuan et al, 2018). Moreover, several studies have demonstrated that mutant KRAS also drives metabolic adaptations in cancers (Son et al, 2013; Kim et al, 2020; Ying et al, 2012; Kerr et al, 2016) and that KRAS-driven metabolic reprogramming depends on the tissue of origin (Mayers et al, 2016; Gwinn et al, 2018), KRAS copy number (Kerr et al, 2016) and mutation (Varshavi et al, 2020), also reviewed in Kerk et al (2021).

[1]MRC Cancer Unit, University of Cambridge, Cambridge CB2 0XZ, UK. [2]Signalling Programme, The Babraham Institute, Babraham Research Campus, Cambridge CB22 3AT, UK. [3]Department of Molecular and Clinical Cancer Medicine Institute of Systems, Molecular and Integrative Biology, University of Liverpool, Liverpool, UK. [4]Cancer Science Institute of Singapore, Singapore 117599, Singapore. [5]NUS Centre for Cancer Research (N2CR), National University of Singapore, Singapore 117599, Singapore. [6]Institute of Molecular and Cell Biology (IMCB), A(∗)STAR, Singapore 138673, Singapore. [7]Department of Oncology, University of Cambridge, Cambridge CB2 0XZ, UK. [8]Department of Medicine, National University of Singapore, Singapore 119228, Singapore. [9]Institute for Metabolomics in Ageing, Cluster of Excellence Cellular Stress Responses in Aging-associated Diseases (CECAD), University of Cologne, Faculty of Medicine and University Hospital Cologne, Cologne, Germany. [10]Institute of Genetics, Faculty of Mathematics and Natural Sciences, Faculty of Medicine, University of Cologne, Cologne, Germany. [11]Centre for Genome Engineering and Maintenance, College of Health, Medicine and Life Sciences, Brunel University London, London UB8 3PH, UK. ✉E-mail: alessandro.esposito@brunel.ac.uk

These observations highlight the importance of understanding the mechanisms underpinning the pathogenicity of specific oncogenic KRAS alleles. Here, we used hotspot mutations at the G12 codon of KRAS to investigate the consequences of different KRAS mutations on cellular phenotype in a panel of colorectal cell lines and to identify common vulnerabilities that could be targeted therapeutically. Transcriptomics and metabolomics revealed remarkable differences and commonalities between G12 mutant cells. More specifically, we identified significant differences in FOXO1 signalling, which regulates glutamine metabolism and provides proliferative advantages to KRAS mutant cells, such as G12D and G12V, under conditions of limited nutrients. The overarching role of KRAS in these metabolic shifts is appreciated (Kerk et al, 2021), but a detailed understanding of how distinct mutations modulate these specific pathways (e.g., glutamine utilisation, de novo synthesis, and nitrogen/ammonia handling) and the regulatory networks involved remains an area of active investigation.

We show that the FOXO1-GLUL axis upregulates glutamine metabolism in these cells, leading to enhanced glutamine synthesis from extracellular glucose. At the same time, the upregulation of FOXO signalling, a pathway so far primarily associated with apoptosis in cancer (Zhao et al, 2010; Kim et al, 2007; Paik et al, 2007; Zhang et al, 2011; Xie et al, 2012), enhances ammonia recycling via glutamine synthesis and transamination pathways, supporting the survival advantage of G12D and G12V mutant cells. Notably, we identified that the simultaneous targeting of glutamine synthesis and glutaminolysis selectively kills G12-mutant KRAS cell lines in two heterozygous isogenic colorectal cell lines (SW48 and LIM1215) compared to their wild-type counterpart, suggesting a high dependency of KRAS mutant cells on nitrogen recycling and a possible new venue for therapeutic intervention.

# Results

### Validation of the SW48 isogenic panel

To investigate the effects of different KRAS G12 mutants, we first characterised the SW48 isogenic colorectal cancer cell line harbouring heterozygous mutations in KRAS at codon 12 (SW48$^{+/+}$, SW48$^{+/G12A}$, SW48$^{+/G12C}$, SW48$^{+/G12D}$, SW48$^{+/G12V}$, hereafter also referred to as WT, G12A/C/D/V). We selected the G12D and G12V mutations because of their high frequency across all KRAS-driven cancers (including colorectal cancer), G12C for its high prevalence only in lung adenocarcinoma, and G12A because it is infrequently observed in colorectal adenocarcinoma yet biochemically indistinguishable from G12V (Hunter et al, 2015a). Immunoprecipitation of GTP-bound KRAS confirmed that the mutant cell lines exhibit upregulated active KRAS (Appendix Fig. S1A). These substantial differences did not translate into evident upregulation of the well-characterised MAPK (mitogen-activated protein kinases) and PI3K (phosphoinositide 3-kinase) effector pathways as assessed by the phosphorylation of the ERK (extracellular signal-regulated kinase) and AKT kinases, respectively (Appendix Fig. S1B). Lack of hyper-activation of these pathways is congruent with prior observation both in cell lines and tumours, particularly in the context of heterozygote mutations (Li et al, 2018b; Konishi et al, 2007; Hood et al, 2019). A modest

upregulation of the ERK pathway was more apparent in serum-starved, low-nutrient conditions (1% FCS, 2 mM Glucose) for the G12V and G12D cell lines (Appendix Fig. S1C), suggesting that these mutants are more capable of sustaining this signalling cascade than the others in limited nutrient and growth factor conditions. We further validated the response of the cell lines using MRTX1133, a potent and selective inhibitor of the KRAS G12D mutant protein; Appendix Fig. S1D shows that MRTX1133 (100-400 nM) significantly decreases the phosphorylation of ERK and AKT in the mutant SW48$^{+/G12D}$ but not in the parental cell line. Moreover, we confirm that MRTX1133 (100 nM) is very specific in inhibiting the KRAS G12D mutant as it has no apparent impact on cells other than SW48$^{+/G12D}$ on the phosphorylation of ERK and AKT when tested on the full SW48 panel (Appendix Fig. S1E).

We also characterised the panel by RNA sequencing to investigate the broader impact of the different KRAS mutant alleles. We found that about 2000 genes are differentially regulated (false discovery rate less than 5% and log-2 fold change larger than 1) in at least one mutant cell line compared to parental SW48 cells (Fig. 1A; Appendix Fig. S1F,G). Gene enrichment analysis (Appendix Tables S1 and 2) highlighted differences in extracellular matrix receptor interactions (e.g., laminins and integrins) and cell migration (semaphorin/plexin signalling) gene sets. Just around one hundred of these genes are similarly upregulated or down-regulated in all mutant cell lines (Appendix Fig. S1H,I) and relate to transcriptional misregulation in cancer, including MAPK and mTOR signalling amongst other enriched KEGG pathways. DCLK1, MET and AKAP12 are some of the genes with significant changes in several mutant lines that have been previously reported (Hammond et al, 2015a).

### KRAS missense mutations at glycine-12 perturb signalling and metabolic pathways

Most of the genes upregulated in G12D also seem to be upregulated in G12V, although to a lesser extent. Given that G12D and G12V are the most prevalent KRAS mutations in colorectal adenocarcinoma, we further analysed the differentially regulated genes in both mutant cell lines (Fig. 1B,C; Appendix Fig. S1F,J, and Appendix Tables S1 and 2). We identified around 300 genes enriched in key metabolic pathways (e.g., ALDH4A1, ALDH5A1, GAD1, GLUL and ABAT) signalling (e.g., IL1R2, MET, SEMA6C, PPP3CA, EFNB3, ABLIM3, PLXNA2, SLIT1, MYL9, MET, NFATC4), and transcriptional regulation (e.g., MEF2C, CEBPB, MYCN, BCL6, FOXO1, DUSP6). Notably, both PCA and gene enrichment analysis of our transcriptomic data (Fig. 1B,C) revealed a significant FOXO signalling signature associated with the G12D and G12V mutants, including the upregulation of FOXO1 itself, alongside other known FOXO target genes and pathway components (see also Appendix Tables S1 and 2), indicative of an enhanced FOXO1-driven transcriptional programme in these cells.

Since we observed significant changes in transcriptional and metabolic signatures, we verified whether these differences are reflected in mutant-specific alterations of cellular metabolism by characterising the mitochondrial function of SW48 cell lines (Fig. 1D). The oxygen consumption rate (OCR) analysis revealed a lower basal and maximal respiration in G12D and G12V mutant cell lines. The extracellular acidification rate (ECAR) of G12D and G12V cells was lower or comparable to wild-type KRAS cells and

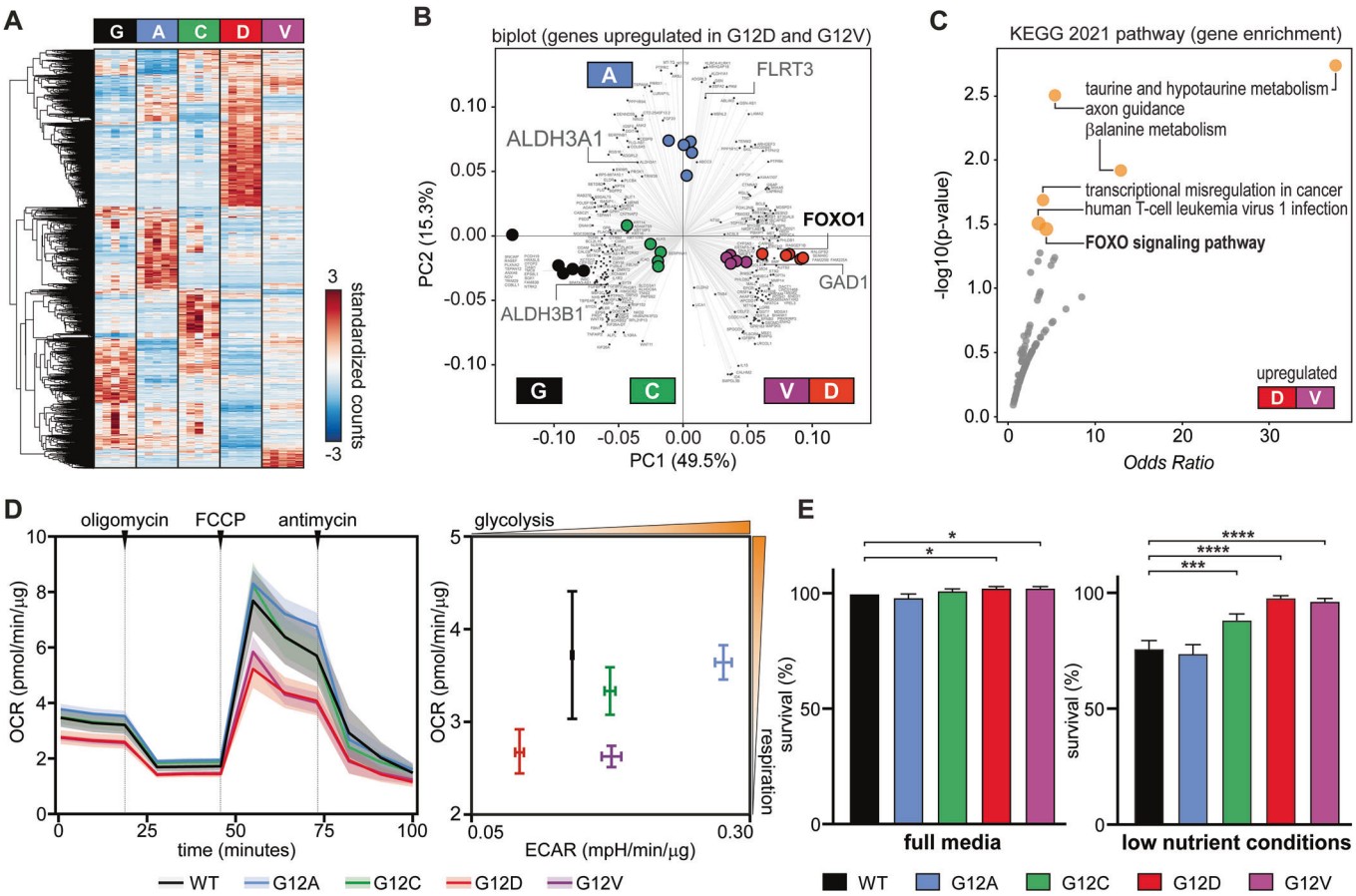

**Figure 1. Isogenic SW48 cell lines reveal common and distinctive features of mutant KRAS alleles.**

(A) Hierarchical clustering showing genes that are differentially expressed in colorectal SW48 cancer cell lines harbouring heterozygous mutations (G12A, G12C, G12D and G12V) at glycine-12 of KRAS ($n = 5$, FDR < 5%, fold change >2). Read counts were standardised for each gene, and each gene shown should be at least differentially regulated in one mutant line. Each cell line exhibits distinctive signatures. This experiment was carried out in full media. (B) Biplots related to PCA analysis with the gene loading in the background and the sample scores coloured in the foreground. The selected genes are upregulated in SW48 G12D and G12V (see also Appendix Fig. S1F–J and Appendix Table S1). Some names of non-coding transcripts are omitted for clarity. (C) Example of gene enrichment analysis using EnrichR and KEGG 2021 pathways to analyse the genes indicated by the orange cluster in panel (Appendix Fig. S1J), suggesting that the G12D and G12V transcriptional signatures are related to metabolism and FOXO signalling. (D) Profiling of the oxygen consumption rate (OCR, left panel) and extracellular acidification rate (ECAR, right panel) obtained by the Seahorse extracellular flux analyser. OCR and ECAR were normalised to total protein content. Shaded areas are 95% confidence intervals, and error bars are standard errors, evaluated over four independent repeats carried out in full media. (E) The growth rate of SW48 isogenic cells is similar in full media (left panel) but differs significantly in low-nutrient, glucose and serum-reduced media conditions (right panel). Data represents an average of three biological replicates, where each graph is normalised to WT in full media. Data are shown as means ± standard deviations. Statistical analyses were performed using one-way ANOVA followed by Dunnett's multiple comparisons test. Only statistically significant $p$ values are shown (*$p = 0.03$, *$p = 0.03$ (full media); ***$p = 0.0002$, ****$p \leq 0.0001$ (low nutrient)). Source data are available online for this figure.

the other mutants, suggesting that glycolysis does not compensate for the lower mitochondrial function of these cells. Differences in cellular respiration did not result in notable differences in cell viability as assessed by sulphorodamine B (SRB) colorimetric assay (Fig. 1E, left panel) under standard culture conditions (RPMI with 11 mM Glucose and 10% FCS). Recognising that standard cell culture media often contain supraphysiological nutrient levels, we performed key functional assays under low nutrient conditions (2 mM glucose and 1% FCS). This glucose concentration, while not modelling severe hypoglycaemia, is substantially reduced from standard RPMI and falls within the range of glucose levels (0.2–2.5 mM) reported in various tumour microenvironments (Nightingale et al, 2019) which are often considerably lower (~5 mM) than plasma levels (Sullivan et al, 2019). This reduction,

coupled with lowered serum, was intended to impose a relevant metabolic stress to unmask differential adaptations and vulnerabilities. In these low-nutrient conditions, wild-type SW48 cells and the G12A line exhibited a substantial reduction in growth. In contrast, G12D and G12V cells exhibited high resilience to the change in nutrient conditions (Fig. 1E, right panel).

## G12D and G12V KRAS mutants boost glutamine synthesis from glucose via FOXO1

To shed light on this unexpected metabolic behaviour of the G12D and G12V mutants, we traced glucose utilisation with $^{13}$C-glucose isotope and liquid chromatography-mass spectroscopy (LC-MS) in low nutrient conditions. Intracellular metabolomics

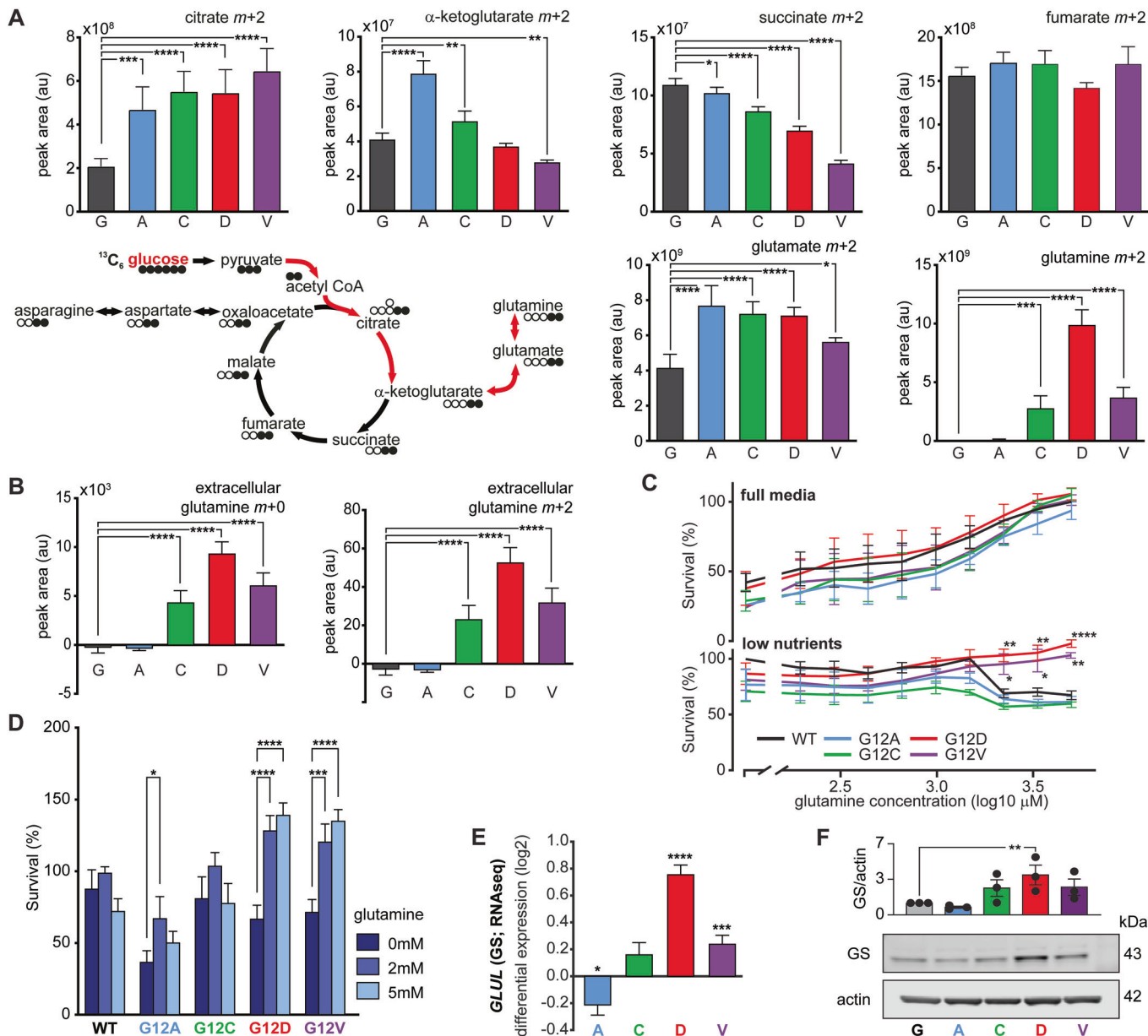

showed that the fluxes of $^{13}$C-glucose carbon in the TCA (tricarboxylic acid) cycle are different across mutant cell lines (Fig. 2A). For example, citrate $m + 2$ is higher than wild-type in all mutant cell lines, fuelling their de novo glutamate production. However, while the G12D, G12V and G12C direct $^{13}$C-glucose carbon flux towards glutamine synthesis, the G12A mutant maintains a higher flux through the TCA cycle and high glutamate levels. The analysis of extracellular metabolites indicated that glutamine $m + 2$ is also released in the media (Fig. 2B). These results indicate that some KRAS mutants can generate glutamine from glucose.

The synthesis of glutamine by some of the mutant cell lines prompted us to test the dependency of mutants on glutamine. In full growth media (10% FCS and 11 mM glucose), all SW48 cells of the isogenic panel increased viability with higher glutamine

concentrations (Fig. 2C, top panel). However, in low nutrient conditions (1% FCS and 2 mM glucose), the wild-type, G12A and G12C cell lines exhibited a significant decline in viability at a glutamine concentration above 1 mM (Fig. 2C, bottom panel). On the contrary, the G12D and G12V mutant cells showed a further increase in viability at high concentrations of glutamine (Fig. 2C,D), thus resulting in a fitness advantage compared to the wild-type cells and the other mutants in nutrient conditions (i.e., low glucose and high glutamine) reminiscent of the tumour microenvironment (Cruzat et al, 2018; Sullivan et al, 2019). qPCR and western blotting (Fig. 2E,F) reveal that glutamine synthetase (GS or *GLUL*) is expressed at a higher level in SW48$^{+/G12D}$ and—to a lower extent—also in the G12V and G12C mutant cell lines. This observation is consistent with the increased synthesis of glutamine measured with metabolomics (Fig. 2A,B).

**Figure 2. Oncogenic KRAS-allele dependent rewiring of cellular metabolism highlights differences in glutamine metabolism in low-nutrient conditions.**

(A) Metabolic flux analysis using $^{13}$C-labelled glucose ($n = 5$ technical replicates). The diagram shows metabolite labelling patterns in the TCA cycle and glutamine pathway, depicting the incorporation of heavy carbon isotopes with full black circles. The red arrows highlight how the TCA cycle incorporates carbon from glucose into glutamine. The graphs show the intracellular abundance of citrate, α-ketoglutaric acid (αKG), glutamate, glutamine, succinate, and fumarate incorporating carbon derived from labelled glucose. Data are shown as means ± standard deviations. Statistical analyses were performed using one-way ANOVA followed by Dunnett's multiple comparisons test. Only statistically significant $p$ values are shown (citrate m + 2 ***$p$ = 0.0009, ****$p$ ≤ 0.0001, αKG m + 2 ****$p$ ≤ 0.0001, **$p$ = 0.003, **$p$ = 0.0025, succinate m + 2 *$p$ = 0.038, ****$p$ ≤ 0.0001, glutamate m + 2 ****$p$ ≤ 0.0001, *$p$ = 0.015, glutamine m + 2 ***$p$ = 0.0001, ****$p$ ≤ 0.0001). (B) Extracellular abundance of unlabelled (m + 0) and labelled glutamine (m + 2) determined by consumption-release measurements ($n = 5$ technical replicates). Data are shown as means ± standard deviations. Statistical analyses were performed using one-way ANOVA followed by Dunnett's multiple comparisons test. Only statistically significant $p$ values are shown (****$p$ ≤ 0.0001). (C) Survival of isogenic SW48 cell lines at varying concentrations of glutamine in full media (up) and low nutrient media conditions (down) measured by SRB assay. Survival curves are the average of 3 biological replicates. The data is normalised to the highest viability of SW48 WT in each media condition. Data are shown as means ± standard errors of the mean. Statistical analyses were performed using two-way ANOVA followed by Dunnett's multiple comparisons test. Only statistically significant $p$ values are shown (G12D **$p$ = 0.003, **$p$ = 0.0019, ****$p$ ≤ 0.0001, G12V *$p$ = 0.03, *$p$ = 0.017, **$p$ = 0.0015). (D) Survival of SW48 isogenic cells in 0, 2 and 5 mM glutamine in low nutrient media conditions. The bar graph is representative of 4 biological replicates shown as means ± standard errors of the mean. Values are normalised to the viability of WT at 2 mM glutamine in each experiment. Statistical analyses were performed using two-way ANOVA followed by Dunnett's multiple comparisons test. Only statistically significant $p$ values are shown (G12A *$p$ = 0.042, G12D ****$p$ ≤ 0.0001, G12V ***$p$ = 0.0006, ****$p$ ≤ 0.0001). (E) Differential expression of the *GLUL* gene coding for glutamine synthetase (GS) as measured by RNAseq and shown with standard errors ($n = 5$ technical replicates; *$p$ = 0.03, ****$p$ ≤ 0.0001, ***$p$ = 0.003). (F) Representative immunoblot showing glutamine synthetase (GS) levels in SW48 cell panel. Graph is a means ± standard deviations of three biological replicates. Statistical analyses were performed using one-way ANOVA followed by Dunnett's multiple comparisons test comparing all mutants to the WT. Only statistically significant $p$ values are shown (G12D *$p$ = 0.01). Source data are available online for this figure.

Interestingly, *GLUL* is known to be under the transcriptional control of FOXO1 (Van Der Vos et al, 2012) and our transcriptomics analysis highlighted FOXO signalling as altered in the G12D and G12V mutant SW48 lines particularly (Fig. 1B,C; upregulation of *BCL6*, *FOXO1* and *FOXO32*). We validated the higher expression of FOXO1 and nuclear localisation by western blotting, qPCR, immunofluorescence and cell fractionation (Fig. 3A–F and Appendix Fig. S3A). An increase of cytoplasmic FOXO1 was detectable in all mutant cells, although nuclear FOXO1 was upregulated more strongly in the G12D followed by G12V mutant lines (Fig. 3F).

To test that the higher expression of FOXO1 in mutant cells depends on the oncogenic KRAS allele, we treated SW48 isogenic panel with the specific KRAS G12D inhibitor MRTX1133 (100 nM). Inhibition of the KRAS G12D oncogenic allele decreases not only phospho-ERK and phospho-AKT levels in G12D mutants, but also decreases AKAP12, a protein overexpressed particularly in KRAS G12D (Appendix Fig. S1D,E) and FOXO1 specifically in the SW48$^{+/G12D}$ cell line (Fig. 3C,D; Appendix Fig. S3B,F), confirming that these G12D-associated phenotypes are a direct consequence of the oncogenic allele rather than non-specific clonal variation within this specific cell line. Interestingly, glutamine synthetase is also downregulated in SW48$^{+/G12D}$ specifically in low nutrient conditions (Fig. 3D) and its regulation via FOXO1 is confirmed in these cells with FOXO1 knock-down (Appendix Fig. S3E).

To test the hypothesis that the metabolic phenotype we observed depends on FOXO1, we quantified gene expression and metabolic flux in these two mutant lines relative to the parental control when treated with the small molecule inhibitor AS1842856 (iFOXO). qPCR analysis (Fig. 3G) confirmed that FOXO1 inhibition down-regulates *GLUL* expression in full nutrient conditions. Furthermore, transcriptomics (Appendix Fig. S3C, and Appendix Tables S2 and 3) permitted us to verify that FOXO1 inhibition downregulates several FOXO1 transcriptional targets and FOXO signalling more generally in addition to *GLUL* expression. We also repeated $^{13}$C-glucose flux analysis in the presence of FOXO1 inhibition, demonstrating that the upregulation of glutamine synthesis from glucose (Fig. 3H) in the G12D and G12V mutant lines depends on FOXO1. Taken together, this data indicates that

FOXO1 is a critical factor in the KRAS-dependent rewiring of glutamine metabolism.

## FOXO1 drives differential nitrogen recycling in G12D and G12V KRAS mutants

The synthesis of glutamine from glutamate is a major mechanism for ammonia detoxification, and FOXO1 has also been linked to nitrogen metabolism and ammonia assimilation (Kamei et al, 2014). Indeed, in addition to glutamine production, in our metabolomics experiment, FOXO1 inhibition significantly affected pathways related to ammonia recycling and nitrogen metabolism, such as the urea cycle, glutamate metabolism, arginine and proline metabolism, glycine and serine metabolism (Appendix Fig. S3C). These results led us to investigate whether ammonia is recycled differently in the mutant lines (G12D and G12V) that exhibit high FOXO1 expression and increased glutamine synthesis.

To monitor ammonia incorporation, we cultured cells with labelled ammonia ($^{15}$NH$_3$ at 3 mM), confirming a significant de novo production of glutamine ($N + 1$ and $N + 2$ isotopologues) incorporating nitrogen from ammonia in G12V and G12D mutant cells (Fig. 4A, first two graphs). Again, both mutant cells export more glutamine to the extracellular space compared to the parental line (Fig. 4A, middle graph); however, G12D cells also incorporated the nitrogen derived from ammonia into several amino acids like aspartate, asparagine, serine, glycine and alanine (Fig. 4B) via transamination reactions. The G12V cells preferentially produce and export glutamine $N + 1$ and $N + 2$ (Fig. 4A) without upregulating transamination reactions. These cells also appear to accumulate aspartate ($N + 1$) due to a lack of asparagine synthesis (Fig. 4B). In addition, G12V showed enhanced ammonia incorporation into hexosamine biosynthesis (Appendix Fig. S4A). We also noted that KRAS mutant cells substantially upregulated the synthesis of ophthalmic acid compared to the parental line. G12V cells were particularly efficient in incorporating labelled nitrogen from free ammonia into ophthalamate (Appendix Fig. S4B).

We complemented these experiments by labelling the amide and alpha nitrogens of glutamine. Efficient incorporation of nitrogen into aspartate, glycine, serine, and alanine in the G12D mutant and

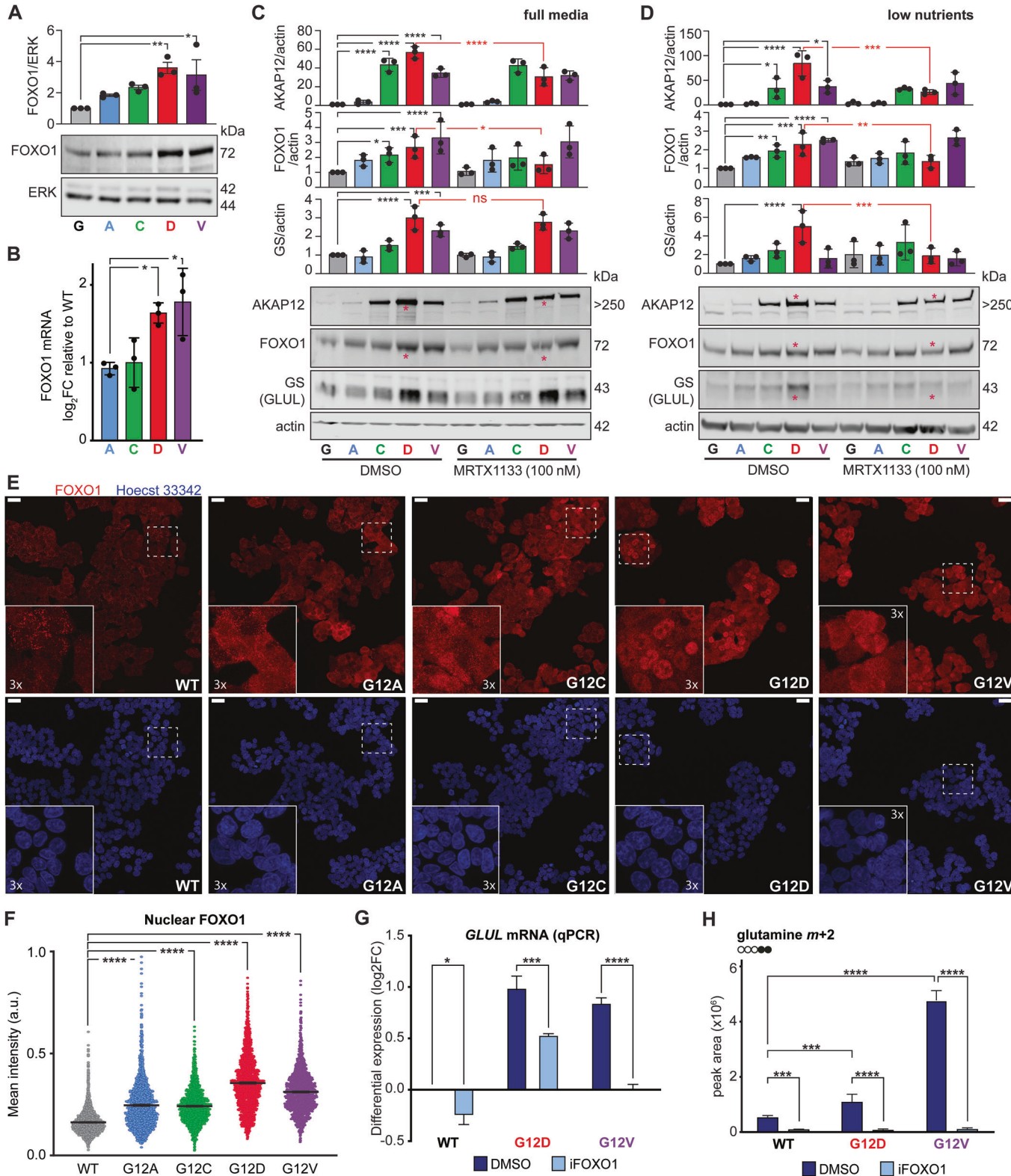

**Figure 3. FOXO1 regulates glutamine synthesis upregulation in KRAS mutant cells.**

(A) Representative immunoblot showing increased expression of FOXO1 in G12D and G12V mutant SW48 cells in full media. Graph is a means ± standard deviations of three biological replicates. Statistical analyses were performed using one-way ANOVA followed by Dunnett's multiple comparisons test comparing all mutants to the WT. Only statistically significant $p$ values are shown (G12D **$p = 0.009$, G12V *$p = 0.024$). (B) mRNA levels of *FOXO1* detected by qPCR. Data shows means ± standard deviations log2FC change of 3 biological repeats in full media. Statistical analyses were performed using one-way ANOVA followed by Dunnett's multiple comparisons test comparing all mutants to the WT. Only statistically significant $p$ values are shown (G12D *$p = 0.03$, G12V *$p = 0.014$). (C,D) Representative immunoblots showing the increased expression of FOXO1 in G12D and G12V mutant SW48 cells in full media (C) or low nutrient conditions (D) with MRTX1133 (100 nM) compared to matched DMSO controls. MRTX1133 decreases the expression of AKAP12, FOXO1 only in SW48$^{+/G12D}$ cells. GS expression is altered but only in low-nutrient media. Red asterisks mark the relevant G12D lanes. Quantifications of western blots are shown as means ± standard deviations of three biological replicates. Statistical analyses were performed using one-way ANOVA followed by Dunnett's multiple comparisons test for all comparisons. Only statistically significant $p$ values are shown (C AKAP12 ****$p \leq 0.0001$; FOXO1 *$p = 0.012$, ***$p = 0.0004$, G12D *$p = 0.012$; GS ****$p \leq 0.0001$, ***$p = 0.0004$; D AKAP12 G12C *$p = 0.0279$, ****$p \leq 0.0001$, G12V *$p = 0.014$, ***$p = 0.0001$; FOXO1 G12C **$p = 0.004$, ***$p = 0.0001$, ****$p \leq 0.0001$, G12D **$p = 0.004$; GS ****$p \leq 0.0001$, ***$p = 0.0001$). (E, F) Representative immunostaining of FOXO1 (red) and nuclear staining (blue) in SW48 isogenic panel in low-nutrient conditions. Scale bar: 25 μm; inserts: 3x magnification. (F) Quantification of mean nuclear intensity (related to E) and standard errors (cells per sample ≥500, $n = 3$ biological repeats). Statistical analyses were performed using one-way ANOVA followed by Dunnett's multiple comparisons test comparing all mutants to the WT. Only statistically significant $p$ values are shown (****$p \leq 0.0001$). (G) qPCR analysis of *GLUL* mRNA levels upon FOXO1 inhibition in full media. Values were normalised to SW48 WT DMSO of each experiment and shown as mean with standard errors. Statistical analyses from 3 biological replicates were performed using two-way ANOVA followed by Sidak's multiple comparisons test for all comparisons. Only statistically significant $p$ values are shown (*$p = 0.0424$, ***$p = 0.0002$, ****$p \leq 0.0001$). (H) Changes in abundance of glutamine ($m + 2$) upon inhibition of FOXO1 in a $^{13}$C-glucose labelling experiment (low nutrient). Data represents the mean of 5 technical repeats and standard deviations. Statistical analyses were performed using two-way ANOVA followed by Sidak's multiple comparisons test for all comparisons. Only statistically significant $p$ values are shown (***$p = 0.0001$, ****$p \leq 0.0001$). Source data are available online for this figure.

ophthalmic acid in both G12D and G12V mutants was seen predominantly from the alpha nitrogen of glutamine, suggesting an involvement of glutamate in these reactions (Appendix Fig. S4C). In low-nutrient conditions, RNAseq analysis also shows that genes within the aspartate, alanine and glutamate metabolism pathway (KEGG2021) are differentially regulated in the mutant lines (Fig. 4C, left panel). We repeated transcriptomics in the most different mutants (G12D and G12V) in the presence of FOXO1 inhibition or a DMSO control (Fig. 4C, right panel). Correlation analysis between log2-fold changes shown in Fig. 4C suggests that most changes in this metabolic pathway might be attributed to the expression of the oncogenic KRAS alleles and alterations mediated by FOXO1 (Fig. 4D).

Metabolomics and transcriptomics thus suggest that KRAS G12D and G12V alleles upregulate nitrogen recycling pathways via FOXO signalling. To test this hypothesis, we repeated the metabolomic analysis of wild-type, G12D, and G12V mutant cells using labelled ammonia and the FOXO1 inhibitor. FOXO1 inhibition resulted in a substantial decrease in glutamine synthesis and transamination reactions (Fig. 4E). Several independent lines of evidence thus suggest that oncogenic KRAS G12D and G12V lead to a higher expression of nuclear FOXO1, which in turn results in increased glutamine synthesis, ammonia detoxification via glutamine synthetase upregulation, and amino acid synthesis via transamination reactions in G12D. These mechanisms could maintain the proliferation potential of these mutant cells even in nutrient-limiting conditions (Fig. 4F) and might thus represent valuable therapeutic targets.

## Combinatorial drugging of glutamine metabolic pathway as a therapeutic strategy

Therefore, we tested if inhibition of FOXO1 in low nutrient conditions would kill mutant cell lines selectively. However, FOXO1 inhibition reduced the viability of all tested cell lines, particularly cells with lower nuclear FOXO1, including KRAS wild-type cells (Appendix Fig. S5A). Therefore, we investigated if targeting glutamine metabolism downstream of FOXO1 could offer

selectivity for mutant cells (Fig. 5C). First, we targeted the glutamine synthesis pathway with the well-characterised GLUL inhibitor methionine sulfoximine or MSO (Rowe et al, 1969; Demarco et al, 1999; Ghoddoussi et al, 2010) without observing substantial differences between KRAS mutant cell lines (Fig. 5A, left panel).

All mutant cell lines exhibit high concentrations of glutamate, which could shift the balance of glutamine anabolism. It is thus conceivable that mutant cell lines might be more sensitive to the inhibition of glutaminase (GLS), the enzyme that converts glutamine to glutamate. We thus tested the effects of CB839, a GLS inhibitor that is already in early clinical trials in kidney and breast cancer (Gross et al, 2014; Raczka and Reynolds, 2019), on the SW48 isogenic panel. GLS inhibition does show some selectivity in the viability of mutant cell lines at the tested concentrations, particularly in G12A and G12D cells (Fig. 5B, left panel). Notably, GLS is downregulated in all mutant lines (Fig. 4C), particularly in the G12A mutant, which also exhibits the highest glutamate concentration within the SW48 panel.

Therefore, we tested whether MSO and CB839, together, could further sensitise the mutant cell lines by targeting features common to all mutant cells (low *GLS* and high glutamate) and more specific to G12D and G12V mutant cells (high glutamine and high *GLUL*). When we titrated MSO and CB839 in the presence of sublethal concentrations of CB839 (100 nM) and MSO (2 mM), respectively, we observed a substantial reduction of the viability of KRAS mutant cells, including the most resilient mutants G12D and G12V (right panels of Fig. 5A,B). Knocking down the two genes by siRNA also confirmed the sensitivity of G12D and G12V mutant cells to the inhibition of GLS and GS together (Appendix Fig. S5C). The titration of both drugs reveals a strong synergistic effect of the GS and GLS inhibitors at all concentrations (Appendix Fig. S5B), particularly in G12D and G12V mutant cell lines, as tested by Bliss analysis (Zheng et al, 2022).

Finally, we investigated how robust our observations are in relation to nutrient conditions and cell types. We repeated the drug-sensitivity experiments at a fixed dose of CB839 (100 nM) and MSO (2 mM) in low nutrient media (1% FCS and 2 mM glucose)

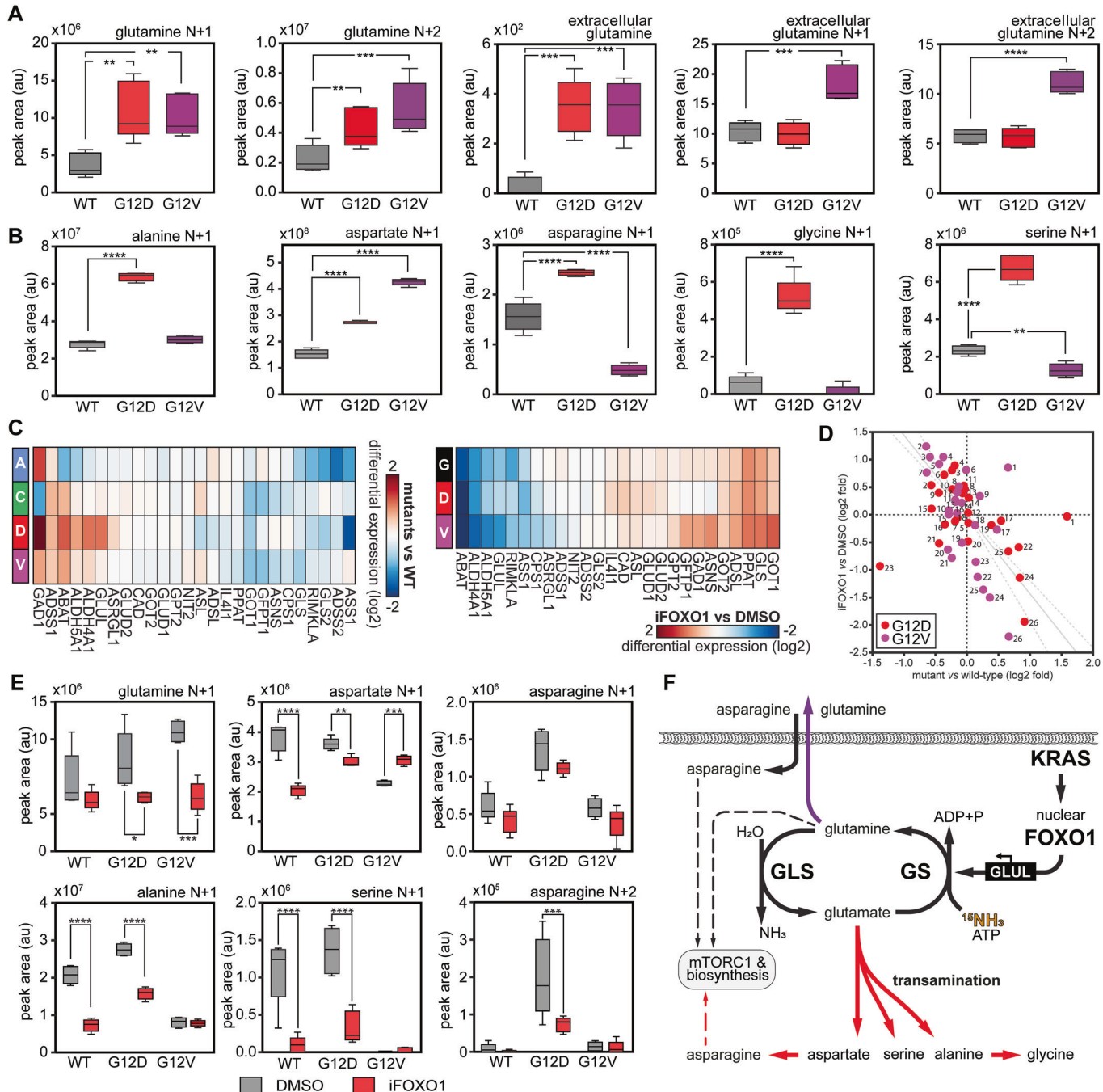

and full RPMI growth media conditions (Fig. 5D). The viability data confirms that cells harbouring the mutant KRAS alleles are substantially more sensitive to the two drugs than the wild-type SW48 cell line. This sensitivity, however, disappeared once glucose and serum concentrations were restored to higher levels, suggesting that additional specificity could be achieved thanks to the tumour microenvironment that is usually deprived of glucose.

To mitigate the risk that the observed sensitivities entirely depend on the SW48 cell line background, we repeated crucial experiments using a second isogenic panel derived from the colorectal cancer cell line LIM1215 (Appendix Fig. S6). The LIM1215 panel harboured the same mutant alleles and exhibited

upregulation of FOXO1. Specific patterns of FOXO1 expression and nuclear localisation showed some variation between the two cell line backgrounds (e.g., Appendix Fig. S6C,D compared to Fig. 3E,F, Appendix Fig. S3A), with the G12D clones exhibiting a stronger phenotype. The LIM1215 mutant cells revealed a conserved vulnerability to the combinatorial inhibition of glutamine synthetase and glutaminase, similar to that observed in the SW48 isogenic panel. Most importantly, the clones that exhibit higher expression of the FOXO1 protein do exhibit enhanced synergy between the drugs. This suggests that the reliance on the glutamine-glutamate cycle might be a common feature of KRAS mutant cells in different colorectal cancer contexts.

◀

**Figure 4.  FOXO1 regulates differential nitrogen recycling in KRAS mutant cells (low nutrient condition).**

(A) Metabolic flux analysis using $^{15}$N-ammonia labelling shows elevated free ammonia incorporation into glutamine in G12D and G12V mutant cells. Synthesised $^{15}$N-labelled glutamine is released at higher rates in extracellular space in G12V mutant. Box plots show the median (centre line), interquartile range (box), and whiskers extending to the minimum and maximum values. Statistical analyses from 5 technical repeats were performed using one-way ANOVA followed by Dunnett's multiple comparisons test comparing all mutants to the WT. Only statistically significant $p$ values are shown (glutamine N + 1 G12D **$p$ = 0.003, G12V **$p$ = 0.007; glutamine N + 2 **$p$ = 0.004, ***$p$ = 0.0006; extracellular glutamine G12D ***$p$ = 0.0002, G12V ***$p$ = 0.0003; extracellular glutamine N + 1 ***$p$ = 0.0002; extracellular glutamine N + 2 ****$p$ ≤ 0.0001). (B) The abundance of $^{15}$N isotopologues in an ammonia labelling experiment showed increased ammonia incorporation into amino acids synthesis via transamination reactions, predominantly in G12D mutants. Box plots show the median (centre line), interquartile range (box), and whiskers extending to the minimum and maximum values. Statistical analyses from 5 technical repeats were performed using one-way ANOVA followed by Dunnett's multiple comparisons test comparing all mutants to the WT. Only statistically significant $p$ values are shown (alanine N + 1 ****$p$ ≤ 0.0001; aspartate N + 1 ****$p$ ≤ 0.0001; asparagine N + 1 ****$p$ ≤ 0.0001; glycine N + 1 ****$p$ ≤ 0.0001; serine N + 1 ****$p$ ≤ 0.0001, **$p$ = 0.006). (C) Hierarchical clustering of gene differential expression (log-2 fold changes) in low-nutrient conditions showing the aspartate, alanine and glutamate metabolism pathway (KEGG pathway *hsa00250*). Left: SW48 mutant cells compared to wild-type cells; right: FOXO1 inhibition compared to DMSO control. We note that not every pathway gene is not detected in each experiment. (D) Correlation plot of the genes detected in both the experiments shown in (C); correlation coefficient of −0.4 and $p$-value ~ 0.001. Legend: (1) GAD1, (2) GOT1, (3) GLS, (4) PPAT, (5) ASNS, (6) ADSL, (7) GFPT1, (8) GOT2, (9) ASL, (10) IL4I1, (11) CAD, (12) GPT2, (13) GLUD1, (14) GLUD2, (15) GLS2, (16) ADSS2, (17) ADSS1, (18) NIT2, (19) ASRGL1, (20) CPS1, (21) RIMKLA, (22) GLUL, (23) ASS1, (24) ALDH4A1, (25) ALDH5A1, (26) ABAT. The solid and dashed grey lines represent the linear regression and the 95% confidence interval, respectively, with GAD1 and ASS1 outliers removed (correlation of −0.7 $p$-value < 10$^{-7}$). (E) Metabolic flux analysis using $^{15}$N-ammonia labelling with inhibition of FOXO1 reveals that the incorporation of ammonia-derived nitrogen in glutamine synthesis as well as in various transaminations depends on FOXO1. Box plots show the median (centre line), interquartile range (box), and whiskers extending to the minimum and maximum values. Statistical analyses from 5 technical repeats were performed using two-way ANOVA followed by Sidak's multiple comparisons test for all comparisons. Only statistically significant $p$ values are shown (glutamine N + 1 *$p$ = 0.029, ***$p$ = 0.0002; aspartate N + 1 ****$p$ ≤ 0.0001, **$p$ = 0.0024, ***$p$ = 0.0002; asparagine N + 2 ***$p$ = 0.0007; ***; alanine N + 1 ****$p$ ≤ 0.0001; serine N + 1 ****$p$ ≤ 0.0001). (F) Diagram showing the fate of $^{15}$N-labelled ammonia in KRAS mutant cells expressing high nuclear FOXO1. Source data are available online for this figure.

Taken together, our results suggest that KRAS mutant cells, particularly those that exhibit upregulation of FOXO signalling, rely on glutamine synthesis and catabolism to survive in limited glucose conditions. Small-molecule inhibitors of clinical interest may be able to target this mechanism to achieve selective killing of oncogenic KRAS cells.

# Discussion

Despite the vast literature on KRAS, the complex mechanisms underpinning the pathogenesis of specific KRAS mutant alleles have yet to be unravelled (Haigis, 2017). Therefore, we have carried out a systematic quantitative characterisation of four mutant isogenic cell lines, comparing the two most common (G12D and G12V) and two rare (G12A and G12C) oncogenic mutations found in colorectal cancer with the parental SW48 line, which is wild-type for KRAS. Consistent with other studies (Varshavi et al, 2020; Hammond et al, 2015b), our initial characterisation revealed diverse gene expression and metabolic profiles across KRAS mutant SW48 cell lines (Fig. 1) that did not always manifest as differences in viability of these cell lines in full media conditions. Providing supraphysiological concentrations of nutrients, full growth media can mask physiologically relevant phenotypes (Gui et al, 2016; Muir and Vander Heiden, 2018; Muir et al, 2017). Therefore, we utilised low-nutrient media (1% serum, 2 mM glucose) more reflective of aspects of the tumour microenvironment (Sullivan et al, 2019; Nightingale et al, 2019). Under this metabolic stress, SW48$^{+/G12D}$ and SW48$^{+/G12V}$ cells notably exhibited increased resilience and enhanced viability in low glucose/high glutamine environments (Fig. 2), a condition reported, for example, in some KRAS-driven tumour models (Sullivan et al, 2019).

Transcriptomics, metabolomics and validation data demonstrate a key role for FOXO1 signalling in the adaptation of cell lines to metabolic reprogramming driven by certain KRAS alleles, particularly G12D and G12V (Figs. 1, 3 and 4). This link between oncogenic KRAS and FOXO1-mediated changes is, to our

knowledge, a novel finding with key implications. While FOXO proteins are often considered tumour suppressors (Paik et al, 2007; Xie et al, 2012), recent work (Kim et al, 2007; Feng et al, 2011; Trinh et al, 2013; Hornsveld et al, 2018) delineates a context-dependent role. Influenced by post-translational modifications and stress pathways (Greer and Brunet, 2005; Essers et al, 2004; Brunet et al, 2004), the diverse nuclear and cytoplasmic activities of FOXO proteins can shift the balance between tumour suppression and promotion (Van Der Heide et al, 2004; Zhao et al, 2010; Cheng, 2022). In SW48$^{+/G12D}$ and SW48$^{+/G12V}$ mutant cell lines, we found that oncogenic KRAS leads to increased FOXO1 expression and nuclear localisation, correlating with higher expression (Fig. 3; Appendix Fig. S3B,F) of its direct transcriptional target (Kamei et al, 2014) glutamine synthetase (GS, encoded by the *GLUL* gene). Interestingly, FOXO1 is the only gene differentially expressed in G12D and G12V mutant cells of the three FOXO transcription factors (FOXO1, FOXO3 and FOXO4) known to be controlled by AKT and to enhance GLUL expression.

Intriguingly, the G12D and G12V KRAS mutations are the most prevalent in colorectal cancer and display the most pronounced changes compared to the milder G12C and WT-like G12A profiles. Despite this, G12D/V SW48 cells revealed unexpectedly lower mitochondrial respiration and glycolysis than wild-type and other mutants (Fig. 1D), hinting at distinct metabolic adaptations.

$^{13}$C-glucose tracing shows that while all mutant cell lines avidly consume pyruvate to feed the TCA cycle for glutamate production and biosynthesis (Fig. 2A), distinct metabolic patterns are evident. The SW48 G12A cell line uniquely maintained high α-ketoglutarate, glutamate levels and flux through the TCA cycle (Appendix Fig. S2F). In contrast, G12D, G12V (and G12C to a lesser extent) cells appear to channel TCA intermediates towards significant de novo glutamine synthesis, a process dependent on the KRAS-FOXO1-GLUL axis. This allele-associated increase in glutamine production, asparagine uptake, along with enhanced glutamate and aspartate synthesis, supports elevated nucleotide production in G12D, G12V and G12C (Appendix Fig. S2A–D). Such enhanced biosynthetic capacity, particularly in G12D/V cells, correlates with

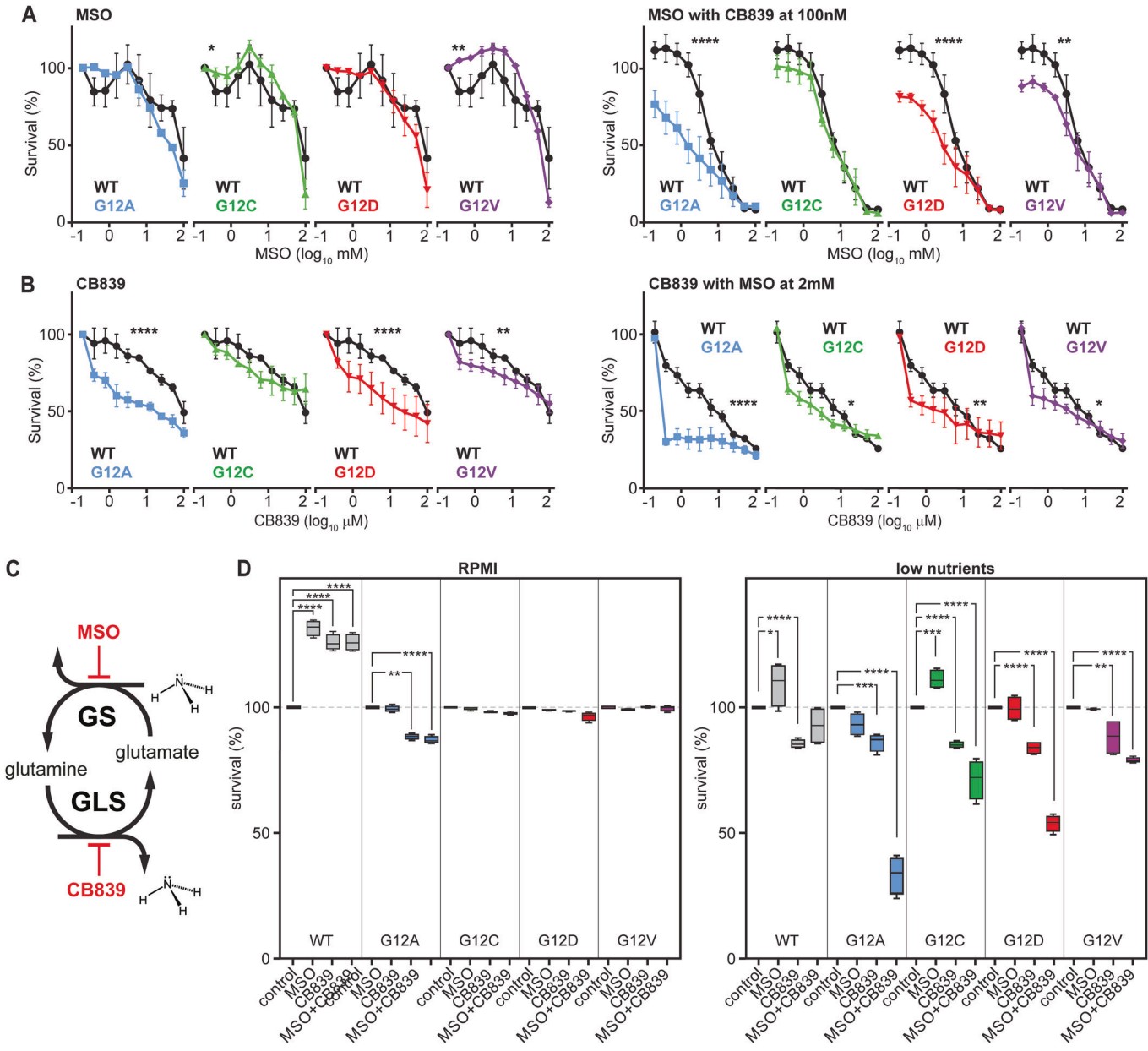

Figure 5. Inhibition of the glutamate-glutamine cycle sensitises KRAS mutant cells.

(A) Viability curves showing titration of GS inhibitor, MSO, alone (left panel) or in combination with sub-lethal doses of GLS inhibitor, CB839 (right panel) in SW48 cells (low nutrient conditions). Data is presented as mean and standard error. Statistical analyses of Area Under the Curve (AUC) from 3 biological repeats were performed using one-way ANOVA followed by Sidak's multiple comparisons test for all comparisons. Only statistically significant $p$ values are shown (left $*p = 0.04$, $**p = 0.005$; right $****p \le 0.0001$, $**p = 0.005$). (B) Viability curves showing titration of the GLS inhibitor, CB839, alone (left panel) or in combination with sub-lethal doses of the GS inhibitor, MSO (right panel), in SW48 cells (low nutrient conditions). Data is presented as mean and standard error. Statistical analyses of Area Under the Curve (AUC) from 3 biological repeats were performed using one-way ANOVA followed by Sidak's multiple comparisons test for all comparisons. Only statistically significant $p$ values are shown (left $****p \le 0.0001$, $**p = 0.0096$) (right $****p \le 0.0001$, $*p = 0.014$, $**p = 0.003$, $*p = 0.025$). (C) Schematic representation of the glutamate-glutamine cycle and the drugs that inhibit the corresponding enzymes in this pathway. (D) Survival graphs of SW48 cells treated with single doses of GLS inhibitor (CB839) 100 nM, GS inhibitor (MSO) 2 mM alone or in combination in full media (left panel) and low nutrient media (right panel) conditions. Data represents the mean of 4 biological replicates. Box plots show the median (centre line), interquartile range (box), and whiskers extending to the minimum and maximum values. Statistical analyses were performed using two-way ANOVA followed by Dunnett's multiple comparisons test for all comparisons. Only statistically significant $p$ values are shown (RPMI WT $****p \le 0.0001$; G12A $**p = 0.01$, $****p \le 0.0001$; Low nutrients WT $*p = 0.014$, $****p \le 0.0001$; G12A $***p = 0.0003$, $****p \le 0.0001$; G12C $***p = 0.0004$, $****p \le 0.0001$; G12D $****p \le 0.0001$; G12V $**p = 0.0015$, $****p \le 0.0001$). Source data are available online for this figure.

increased mTOR pathway activity, as indicated by the phosphorylation of 4EBP1 (EIF4EBP1), linking their specific metabolic reprogramming to proliferative signalling (Appendix Fig. S1B,C).

While G12D and G12V are very similar, they also exhibit significant differences in nitrogen recycling, some of which are regulated by FOXO1. Nitrogen flux analysis using [15]N-labelled ammonia and glutamine (Fig. 4; Appendix Fig. S4) shows, for example, that SW48 G12D cells use high glutamine production to fuel transamination reactions for synthesising other amino acids (alanine, serine, and glycine). Conversely, G12V cells exhibit a much higher production and export of glutamine, greater synthesis of aspartate and a striking increase in ophthalmic acid synthesis (Fig. 4; Appendix Fig. S4). Additionally, the SW48[+/G12V] line uniquely fixed nitrogen through glycogenesis and hexosamine pathways (e.g., in UDP-glucose, GDP-glucose, UDP-N-acetylglucosamine and sialic acid), a nitrogen recycling pathway linked to colorectal cancer metastasis (Jiang et al, 2019; Paneque et al, 2023).

We also observe that G12V and G12D cell lines exhibit higher intracellular ammonia than the parental line (Appendix Fig. S5D). Ammonia has long been considered a metabolic waste product, but several publications have now shown that the metabolic recycling of ammonia supports the proliferation of cancer cells (Kim et al, 2017; Spinelli et al, 2017; Lie et al, 2019; Cheng et al, 2022). For instance, Spinelli and colleagues have shown that free ammonia released through reductive amination by glutamate dehydrogenase (GLUD, also known as GDH) is utilised for amino acid synthesis by breast cancer cells, leading to tumour growth (Spinelli et al, 2017). The enhanced fitness of G12D and G12V cells in low glucose and high glutamine (Fig. 2C,D), fuelled by their increased capacity for de novo glutamine synthesis from glucose (Figs. 2A,B and 3H), highlights the metabolic reprogramming associated with these alleles. Conversely, the decreased viability observed in wild-type, G12A, and G12C cells under these same conditions when glutamine concentrations exceeded 1 mM (Fig. 2C) may reflect their limited capacity to manage the byproducts of high glutamine flux, such as ammonia, when other nutrients are scarce. Unlike G12D/V cells, which upregulate the FOXO1-GLUL axis and demonstrate enhanced ammonia recycling (Figs. 3 and 4), these cell lines might be more susceptible to ammonia toxicity or other metabolic imbalances induced by excessive glutamine catabolism in a nutrient-stressed state.

Our observations suggest that the oncogenic mutants commonly found in colorectal cancer exhibit a similar phenotype, i.e., they can use excess ammonia and adapt nitrogen metabolism to fuel biosynthetic pathways. This phenotype is directly linked to changes in the expression or localisation of FOXO1 downstream of KRAS signalling, supporting the well-established role of FOXO1 in regulating glutamine, nitrogen metabolism and ammonia detoxification (Kamei et al, 2014; Karkoutly et al, 2024) (Fig. 4; Appendix Fig. S4). Moreover, these metabolic changes observed in G12D and G12V SW48 cells show striking similarities to the consensus molecular subtype 3 (CMS3) of colorectal cancer types, which show an association between increased KRAS mutations and elevated metabolic signature of glutamine and nitrogen metabolism (Guinney et al, 2015).

We note that the exchange of metabolites between different cell types can significantly impact cellular viability within tissues in physiological conditions and pathology. Notable examples are the glutamate-glutamine cycle between neurons and glia but also exchanges of glutamine between cancer-associated fibroblasts and cancer cells that support further tumour progression (Li et al, 2021; Mestre-Farrera et al, 2021; Yang et al, 2016). It is thus intriguing to observe that differential regulation of nitrogen metabolism confers resilience to G12D and G12V SW48 clones and, at the same time, alters the exchange of metabolites to and from the extracellular space. Although beyond the scope of our work, these observations raise the possibility that the exchange of metabolites between mutant cells and the tissue microenvironment could favour the emergence of specific mutations in permissive tissues. The involvement of FOXO1 in colorectal cancers and KRAS-induced metabolic reprogramming is not well-characterised. Interestingly, gains and amplifications of *FOXO1* are evident in colorectal cancers (Aaltonen et al, 2020), and levels of *FOXO1* show a moderate increase with advanced tumour stages of colorectal cancer (Appendix Fig. S7). Our results suggest that specific oncogenic KRAS alleles might upregulate FOXO1 expression and promote its nuclear localisation.

The impact that specific KRAS oncogenic alleles have on tumorigenesis and response to treatment has also been demonstrated in vivo by developing mice harbouring different mutations at the G12 and G13 codons and identifying allele- and tissue-specific phenotypes (Zafra et al, 2019; Winters et al, 2017). With a murine model of lung adenocarcinoma, Kerr et al. have also shown that allele dosage induces distinct metabolic phenotypes (Kerr et al, 2016). These observations, together with the notion that KRAS mutations present clinically first as a heterozygote mutation, make using a human isogenic cell line like SW48 extremely relevant. This highlights that the oncogenic potential of a KRAS mutation may not simply be a matter of constitutive activation, but rather how the specific allele engages downstream pathways to hit a 'sweet spot' of signalling (Li et al, 2018a; Haigis et al, 2019; Cook et al, 2021). Such a signal must be potent enough to confer a fitness advantage yet calibrated to avoid strong tumour-suppressive responses, a balance likely dictated by both the allele-intrinsic biochemistry and the tissue and cell type of origin.

Also, the overall levels of active KRAS are important, but our data suggest that a simple dosage model does not fully explain the distinct metabolic phenotypes observed across the G12 mutants. For example, in SW48 cells, KRAS G12V shares key metabolic reprogramming features with G12D, despite potentially different levels of GTP-bound KRAS, whereas G12A, even with comparable active KRAS levels to some other mutants (Appendix Fig. S1A), diverges in its downstream FOXO1 engagement. This aligns with biochemical studies demonstrating intrinsic differences in GTP hydrolysis, effector engagement (Hunter et al, 2015b), and signalling dynamics among KRAS G12 variants (Hunter et al, 2015a; Ihle et al, 2012; Munoz-Maldonado et al, 2019; Yuan et al, 2018; Gillies et al, 2012). These inherent allele-specific biochemical properties likely contribute to the qualitative and quantitative differences in the resulting metabolic landscapes and underpin the observed variations in FOXO1 expression, nuclear localisation, and overall functional state.

Although not in the context of oncogenic KRAS, FOXO1 is known to be phosphorylated by AKT, ERK and PKA (Lee et al, 2011; Asada et al, 2007; Rena et al, 1999). We do not observe significant changes in AKT (phospho-Ser473) activation across mutant lines, but we detect a robust increase in the phosphorylation

of 4EBP1 downstream of the PI3K-AKT-mTOR axis (Appendix Fig. S1B,C). More importantly, G12D and G12V mutant lines exhibit higher phosphorylated ERK under low-nutrient conditions (Appendix Fig. S1C) and a vast increase in the expression of the scaffold protein AKAP12 (also present in G12C to a lesser extent; Appendix Fig. S1B,C), a protein directly implicated in PKA-mediated phosphorylation of FOXO1. While the precise contribution of each pathway and the specific post-translational modifications on FOXO1 were beyond the scope of this study, it is plausible that KRAS-driven alterations in PI3K/AKT/mTOR, ERK and PKA signalling not only modulate FOXO1 functional state but also drive its mRNA levels through FOXO1 transcriptional autoregulation (Shen et al, 2014).

In this work, we reveal a novel role for FOXO signalling, showing the allele-dependent upregulation of FOXO1. We demonstrate how FOXO1 inhibition, or inhibition of KRAS G12D in SW48$^{+/G12D}$, results in the downregulation of glutamine synthetase. We observe that the inhibition of FOXO1 abrogates the KRAS-dependent upregulation of glutamine synthesis and transamination features. Our work suggests that FOXO1 inhibition might be a viable therapeutic route. However, the inhibition of FOXO1 using AS1842856 is toxic for all cell lines we tested, particularly those wild-type for KRAS. We, therefore, targeted glutamine-glutamate metabolism, which is altered in KRAS mutants, using the glutamine synthetase inhibitor methionine sulfoximine (MSO) and the glutaminase inhibitor CB839 to avoid the toxicity that might result from targeting a key transcription factor such as FOXO1.

Individually, MSO and CB839 show no or moderate selectivity towards mutant cells, respectively (Fig. 5; Appendix Fig. S5). This observation is unsurprising mainly because isogenic panels of cell lines engineered from cancer cell lines are notoriously difficult to kill with specificity. However, we demonstrate a strong synergistic interaction between MSO and CB839 in reducing the viability of KRAS mutant cells. This synergy allows for efficacy at concentrations where individual agents are less potent and, crucially, inhibiting selectively growth of KRAS mutant cells over wild-type SW48 cells, particularly under low-nutrient conditions where their metabolic reprogramming is most evident (Fig. 5; Appendix Figs. S5 and S6). The heightened sensitivity of mutant cells is confirmed by siRNA targeting *GLUL* and *GLS*, and in the LIM1215 panels, despite LIM1215 G12V displaying a narrower synergistic window.

Crucially, treatment of cells with the G12D-specific inhibitor MRTX1133 abrogates this vulnerability (Appendix Fig. S5E) in SW48 +/G12D, confirming that the specificity and synergistic effects of MSO and CB839 depend on the heterozygous KRAS G12D allele. This selective effect of the MSO/CB839 combination contrasts sharply with the general toxicity observed with direct pharmacological inhibition of upstream regulators like FOXO1 (using AS1842856), which non-selectively reduced the viability of all cell lines, including parental wild-type cells (Appendix Fig. S5A). The notable sensitivity of G12A cells to dual GS/GLS inhibition, despite their low expression of nuclear FOXO1 and GS, likely stems from their low basal GS activity coupled with a KRAS G12A-driven reliance on GLS for processing exogenous glutamine to meet heightened metabolic demands. This highlights that different KRAS alleles can create vulnerabilities within the glutamine-glutamate cycle through varied mechanisms, broadening the potential of this therapeutic approach beyond just G12D/V mutants.

The differential regulation of glutamine metabolism and its importance in tumorigenesis is well-documented (Gaglio et al, 2011; Son et al, 2013; Ko et al, 2011; Bott et al, 2019), making it the optimal candidate for therapy. Treatment strategies like inhibition of GLUL (Bott et al, 2019), treatment with glutamine analogue DON (Encarnación-Rosado et al, 2024), or using the GLS inhibitor CB839 (Biancur et al, 2017) are promising in PDAC mouse models; however, alternative compensatory mechanisms have often led to the emergence of resistance. Metabolic pathways are very resilient and regulated to provide the appropriate metabolite levels utilising different pathways. Our results suggest that the pharmacological disruption of glutamine-glutamate metabolism at both sides of the glutamine-glutamate cycle is a promising strategy for the selective elimination of KRAS mutant cells. While CB839 is already clinically approved, MSO has limited clinical use because of its convulsive effects in vivo. In the absence, to our knowledge, of alternative inhibitors for glutamine synthetase that are clinically approved, our results indicate the necessity to investigate the use of low doses of MSO in conjunction with therapeutic doses of CB839 or at least the opportunity to direct drug discovery programmes in identifying new GS inhibitors.

It is important, however, to acknowledge certain limitations in this study. Our conclusions are derived from engineered isogenic cell lines in which KRAS mutations were artificially introduced. We recognise that these in vitro models, based on single clones, cannot fully recapitulate the genetic heterogeneity and evolutionary context of tumours harbouring endogenous mutations. To mitigate this, we confirmed our central therapeutic vulnerability in a second, independent colorectal cancer background (LIM1215), demonstrating the phenotype is not a cell-line-specific artefact. Most critically, we established that the key phenotypes are directly dependent on the oncogenic allele itself by using the KRAS G12D-specific inhibitor MRTX1133 to reverse them. Nonetheless, these models cannot fully capture the complexity of three-dimensional tissues or systemic physiology, even when cultured in low-nutrient conditions designed to reflect aspects of the tumour microenvironment.

Despite these limitations, our work suggests that the glutamine-glutamate cycle provides a growth advantage to KRAS mutant cells with a functional role of the KRAS-FOXO1-glutamine metabolism axis. However, the precise upstream mechanisms linking each KRAS G12 variant to the differential regulation of FOXO1 were not exhaustively detailed. Therefore, the contribution of specific post-translational modifications, for example, remains an important area for future investigation. Taken together, our work suggests that the glutamine-glutamate cycle and KRAS-dependent modulation of FOXO signalling are potentially attractive new targets for validation studies in more clinically relevant models and drug discovery.

## Methods

**Reagents and tools table**

| Reagent/Resource | Reference or source | Identifier |
| --- | --- | --- |
| **Antibodies** | | |
| FOXO1 monoclonal antibody | ThermoFisher Scientific | Cat#MA5-14846 |
| AKAP12 antibody | Sigma | Cat#HPA006344 |

| Reagent/Resource | Reference or source | Identifier |
|---|---|---|
| Glutamine synthase antibody | Abcam | Cat#ab73593 |
| Phospho-4E-BP1 (Thr37/46) | Cell Signalling | Cat#2855S |
| Anti-Ras Antibody, clone RAS10 | Merck | Cat#05-516 |
| Phospho-p44/42 MAPK (ERK1/2) | Cell Signalling | Cat#9106 |
| p44/42 MAPK (ERK1/2) | Cell Signalling | Cat#137F5 |
| Phospho-Akt (Ser473) (D9E) | Cell Signalling | Cat#4060 |
| Akt (pan) (40D4) | Cell Signalling | Cat#2920 |
| Anti-ß actin | Merck | Cat#A5441 |
| Histone H3 | Cell Signalling | Cat#9715 |
| Anti-Tubulin | Santa Cruz | Cat#sc-32293 |
| IRDye 680RD Goat anti-Mouse IgG | LI-COR | Cat#926-68070 |
| IRDye 680RD Goat anti-Rabbit IgG | LI-COR | Cat#926-68071 |
| IRDye 800CW Goat anti-Mouse IgG | LI-COR | Cat#926-32210 |
| IRDye 800CW Goat anti-Rabbit IgG | LI-COR | Cat#926-32211 |
| Alexa Fluor 568, Goat anti-Rabbit IgG (H + L) Cross-Adsorbed Secondary Antibody | ThermoFisher Scientific | Cat#A-11-011 |
| Hoechst 33342 | ThermoFisher Scientific | Cat#H3570 |
| HSP90 | BD Biosciences | Cat#610418 |
| **Chemicals, peptides, and recombinant proteins** | | |
| RPMI 1640, GlutaMAX | Gibco | Cat#11554526 |
| Seahorse XF RPMI medium | Agilent | Cat#103576-100 |
| Seahorse XF 1.0 M glucose | Agilent | Cat#103577-100 |
| L-glutamine (200 mM) | ThermoFisher Scientific | Cat#25030-024 |
| Sodium bicarbonate 7.5% solution | ThermoFisher Scientific | Cat#25080-094 |
| Insulin solution human | Sigma | Cat#I9278 |
| Hydrocortisone - water soluble | Sigma | Cat#H0396 |
| Antimycin | Sigma | Cat#A8674 |
| Oligomycin | Sigma | Cat#O4876 |
| Carbonyl cyanide 4-(trifluoromethoxy) phenylhydrazone (FCCP) | Sigma | Cat#C2920 |
| Sulforhodamine B | Sigma | Cat#230162 |
| Trichloroacetic acid, 99.0% (titration) | Sigma | Cat#T4885 |

| Reagent/Resource | Reference or source | Identifier |
|---|---|---|
| LC–MS grade methanol | Fisher Scientific | Cat#10284580 |
| LC–MS grade acetonitrile | Fisher Scientific | Cat#10001334 |
| L-valine (D$_8$, 98%) | CK Isotopes | Cat#DLM-488 |
| D-glucose | CK Isotopes | Cat#CLM-1396-5-PK |
| FOXO1 Inhibitor, AS1842856 | Merck | Cat#344355 |
| L-glutamine (amide-$^{15}$N, 98%+) | CK Isotopes | Cat#NLM-557 |
| L-glutamine (α-$^{15}$N, 98%) | CK Isotopes | Cat#NLM-1016 |
| Ammonium chloride ($^{15}$N, 99%) | CK Isotopes | Cat#NLM-467 |
| L-methionine sulfoximine | Sigma | Cat#M5379 |
| CB839 (Telaglenastat) | Selleckchem | Cat#S7655 |
| DTT | ThermoFisher Scientific | Cat#R0861 |
| BSA | Sigma | Cat#A9418 |
| NuPAGE MOPS SDS running buffer (20X) | ThermoFisher Scientific | Cat#NP0001 |
| NuPAGE transfer buffer (20X) | ThermoFisher Scientific | Cat#NP0006 |
| NuPAGE LDS sample buffer (4X) | ThermoFisher Scientific | Cat#NP0007 |
| cOmplete, mini, EDTA-free protease inhibitor cocktail | Merck | Cat#11836170001 |
| Phosphatase inhibitor cocktail 2 | Merck | Cat#P5726 |
| Phosphatase inhibitor cocktail 3 | Merck | Cat#P0044 |
| Formaldehyde 16% | ThermoFisher Scientific | Cat#11586711 |
| SuperScript III reverse transcriptase | ThermoFisher Scientific | Cat#18080093 |
| SYBR green universal master mix | ThermoFisher Scientific | Cat#4309155 |
| Lipofectamine RNAiMAX transfection reagent | ThermoFisher Scientific | Cat#13778075 |
| Opti-MEM | ThermoFisher Scientific | Cat#31985070 |
| Mowiol 4-88 | Sigma Aldrich | Cat#81381 |
| MRTX1133 | FisherScientific | Cat#18754856 |
| **Critical commercial assays** | | |
| Pierce BCA Protein Assay kit | ThermoFisher Scientific | Cat#23227 |
| Active Ras Pull-Down and Detection kit | ThermoFisher Scientific | Cat#16117 |
| NE-PER™ Nuclear and Cytoplasmic Extraction Reagents | ThermoFisher Scientific | Cat#78833 |

| Reagent/Resource | Reference or source | Identifier |
|---|---|---|
| RNeasy mini kit | Qiagen | Cat#74106 |
| Ammonia Assay kit | Abcam | Cat#ab83360 |
| **Experimental models: cell lines** | | |
| SW48 Parental | Horizon Discovery Ltd. | Cat#HDPAR-006 |
| SW48 KRAS (G12A/+) | Horizon Discovery Ltd. | Cat#HD103-009 |
| SW48 KRAS (G12C/+) | Horizon Discovery Ltd. | Cat#HD103-006 |
| SW48 KRAS (G12D/+) | Horizon Discovery Ltd. | Cat#HD103-011 |
| SW48 KRAS (G12V/+) | Horizon Discovery Ltd. | Cat#HD103-007 |
| LIM1215 Parental | Horizon Discovery Ltd. | Cat#HDPAR-108 |
| LIM1215 KRAS (G12A/+) | Horizon Discovery Ltd. | Cat#HD116-003 |
| LIM1215 KRAS (G12C/+) | Horizon Discovery Ltd. | Cat#HD116-007 |
| LIM1215 KRAS (G12D/+) | Horizon Discovery Ltd. | Cat#HD116-005 |
| LIM1215 KRAS (G12V/+) | Horizon Discovery Ltd. | Cat#HD116-006 |
| **Oligonucleotides** | | |
| Hs_ACTB_2_SG | Qiagen | Cat#QT01680476 |
| Hs_GLUL_1_SG | Qiagen | Cat#QT00085155 |
| Hs_FOXO1_1_SG | Qiagen | Cat#QT00044247 |
| AllStars negative control siRNA | Qiagen | Cat#1027281 |
| Hs_GLS_6 siRNA | Qiagen | Cat#SI031550190 |
| Hs_GLUL_9 siRNA | Qiagen | Cat#SI04332048 |
| Hs_FOXO1 siRNA Flexitube | Qiagen | Cat# GS2308 |
| **Software and algorithms** | | |
| Seahorse Wave Desktop | Agilent | www.agilent.com |
| Prism | GraphPad | www.graphpad.com |
| StepOne | Appled Biosystems | www.thermofisher.com |
| ImageJ/Fiji | | fiji.sc |
| Cell Profiler | Cell Profiler | www.cellprofiler.org |
| Image Studio | LI-COR | www.licor.com |
| SynergyFinder Plus | University of Helsinki | synergyfinder.org |
| Metaboanalyst 6.0 | | metaboanalyst.ca |
| Adobe Illustrator | Adobe | adobe.com |
| EnrichR | Ma'ayan Lab | maayanlab.cloud/Enrichr/ |
| Venn diagrams | University Gent | bioinformatics.psb.ugent.be/webtools/Venn/ |
| cBioPortal | Center for Molecular Oncology, Memorial Sloan Kettering Cancer Center | cbioportal.org |

| Reagent/Resource | Reference or source | Identifier |
|---|---|---|
| STAR (version 2.6.0c) | Dobin et al (2013) | github.com/alexdobin/STAR/ |
| Cutadapt | https://doi.org/10.14806/ej.17.1.200 | cutadapt.readthedocs.io |
| Gencode | | www.gencodegenes.org/human/ |
| Rsubread (v1.28.1) | Liao et al (2019) | https://doi.org/10.18129/B9.bioc.Rsubread |
| DESeq2 (v1.18.1) | Love et al (2014) | https://doi.org/10.18129/B9.bioc.DESeq2 |
| AccuCor algorithm | Su et al (2017) | github.com/lparsons/accucor |
| Tracefinder 5.0 | Thermo Fisher | www.thermofisher.com |
| Matlab (up to R2024a) | Mathworks | www.mathworks.com |
| Custom Matlab code (RNA seq) | this paper | Data sources files of relevant figures |
| **Other** | | |
| TECAN Plate reader | TECAN | |
| BMG PHERAstar plate reader | BMG Labtech | |
| Q Exactive Hybrid Quadrupole-Orbitrap Mass spectrometer (HRMS) | ThermoFisher Scientific | |
| Dionex Ultimate 3000 UHPLC | ThermoFisher Scientific | |
| Leica SP5 Confocal Microscope | Leica Microsystems | |
| LI-COR ODYSSEY CLx scanner | LI-COR | |
| Nunc Lab-Tek Chamber Slide, 8-well | ThermoFisher Scientific | Cat#177402 |
| Immobilon-P | Merck | Cat#IPVH00005 |
| Q Exactive Hybrid Quadrupole-Orbitrap Mass spectrometer (HRMS) coupled to a Dionex Ultimate 3000 UHPLC | Thermo Scientific | |

## Cell culture

SW48 isogenic cells (SW48$^{+/+}$, SW48$^{+/G12A}$, SW48$^{+/G12C}$, SW48$^{+/G12D}$, SW48$^{+/G12V}$), derived from colorectal adenocarcinoma were purchased from Horizon Discovery. Cells were cultured in their recommended conditions, also referred in the text as full media conditions, RPMI 1640 with GlutaMAX, HEPES (Gibco, # 11554526) and 10% foetal bovine serum (FBS; Gibco, #A5256701). LIM1215 isogenic cells (LIM1215$^{+/+}$, LIM1215$^{+/G12A}$, LIM1215$^{+/G12C}$, LIM1215$^{+/G12D}$, LIM1215$^{+/G12V}$), derived from human colorectal carcinoma, were purchased from Horizon Discovery. Cells were cultured in their recommended media, RPMI 1640 with GlutaMAX, HEPES (Gibco, # 11554526) supplemented with 10% FBS, 1 µg/ml insulin (Sigma, I9278), and 1 µg/ml hydrocortisone (Sigma, H0396). Seahorse XF RPMI medium

(Agilent, #103576-100) supplemented with 2 mM Glucose (Agilent, #103577-100), 2 mM glutamine (ThermoFisher ScientificTM, #25030-024), 0.2% sodium bicarbonate (ThermoFisher ScientificTM, #25080-094) and 1% FBS was used for experiments carried out in low nutrient conditions with both cell lines. Cross-contamination of cell lines was checked by periodic STR profiling and genotyping by sequencing with primers provided by Horizon Discovery. Cell cultures were also periodically tested for mycoplasma.

## Seahorse assay

For mitochondrial stress test, Agilent Seahorse XF Assays were performed in a 24-well XF Cell Culture Microplate using Seahorse XFe24 analyser (Agilent Technologies). Briefly, SW48 cells were plated at a density of $2 \times 10^5$ cells/well in 24-well XF microplates (Agilent Technologies) to achieve a monolayer confluency. Next day, 1 h prior to the experiment, the media was replaced with 675 µl/well Seahorse XF RPMI Medium (Agilent Technologies) supplemented with 1% FBS, 11 mM Glucose and 2 mM Glutamine, mimicking the original nutrient concentrations of full media. During the assay, the following compounds were injected sequentially, 75 µl Oligomycin (10 µM) (Sigma), 83 µl FCCP (5 µM) (Sigma), 92 µl Antimycin (10 µM) (Sigma) and Rotenone (10 µM) (Sigma) together, and OCR and ECAR values were measured. At the end of the assay cells were harvested using 30 µl of RIPA buffer/well and total protein amount was measured using Pierce BCA Protein colorimetric assay (ThermoFisher Scientific). OCR and ECAR values were then normalized to total protein levels and graphs plotted using Graphpad.

## Sulforhodamine-B (SRB) viability assay

For SRB assays both in full media and low nutrient conditions, SW48 isogenic cells were initially seeded on a 96-well plate at a $7 \times 10^3$ cells/well density in full media conditions. 24 h after seeding and complete attachment, media was replaced with either full media or low nutrient media. After further 72 h in culture, cells were used for the SRB assay. Briefly, cells were washed once with PBS and fixed with 100 µl/well 1% (v/v) trichloroacetic acid (Sigma) for 2 h at 4 °C. Fixed cells were washed twice with deionized water and stained with 100 µl/well Sulforhodamine B (Sigma) solution (0.057% w/v in 1% acetic acid) for 1 h at room temperature. SRB was then removed, plates washed with 1% acetic acid 3 times and air dried. SRB stain was solubilized with 10 mM Tris base for 10 min. Absorbance was measured at 540 nm using a 96-well plate reader (BMG Pherastar Plus) and results calculated as percentage survival using Graphpad.

## Liquid chromatography coupled to mass spectrometry (LC–MS)

HILIC chromatographic separation of metabolites was achieved using a Millipore Sequant ZIC-pHILIC analytical column (5 µm, $2.1 \times 150$ mm²) equipped with a $2.1 \times 20$ mm² guard column (both 5 mm particle size) with a binary solvent system. Solvent A was 20 mM ammonium carbonate, 0.05% ammonium hydroxide; Solvent B was acetonitrile. The column oven and autosampler tray were held at 40 °C and 4 °C, respectively. The chromatographic gradient was run at a flow rate of 0.200 ml/minute as follows:

0–2 min: 80% B; 2–17 min: linear gradient from 80% B to 20% B; 17–17.1 min: linear gradient from 20% B to 80% B; 17.1–22.5 min: hold at 80% B. Samples were randomised and analysed with LC–MS in a blinded manner with an injection volume was 5 µl. Pooled samples were generated from an equal mixture of all individual samples and analysed interspersed at regular intervals within the sample sequence as quality control.

Metabolites were measured with a Thermo Scientific Q Exactive Hybrid Quadrupole-Orbitrap Mass spectrometer (HRMS) coupled to a Dionex Ultimate 3000 UHPLC. The mass spectrometer was operated in full-scan, polarity-switching mode, with the spray voltage set to +4.5 kV/−3.5 kV, the heated capillary held at 320 °C, and the auxiliary gas heater held at 280 °C. The sheath gas flow was set to 25 units, the auxiliary gas flow to 15 units, and the sweep gas flow was set to 0 units. HRMS data acquisition was performed in a range of $m/z = 70$–$900$, with the resolution set at 70,000, the AGC target at $1 \times 10^6$, and the maximum injection time (Max IT) at 120 ms. Metabolite identities were confirmed using two parameters: (1) precursor ion m/z was matched within 5 ppm of theoretical mass predicted by the chemical formula; (2) the retention time of metabolites was within 5% of the retention time of a purified standard run with the same chromatographic method.

## LC–MS metabolomics: tracing experiments

For the glucose tracing experiment, SW48 isogenic cells were plated on 6-well plates at a density of $8 \times 10^5$ cells/well in full media conditions (5 replicates for each cell line). The next day, after a PBS wash, the media was changed into low nutrient media where 2 mM glucose was substituted with equimolar concentration of ¹³C-labelled glucose (D-glucose, CK Isotopes) and cultured for another 24 h before extraction.

For extraction, as a first step cell numbers was determined for each cell line using a separate plate for counting. Cells were washed twice in PBS and incubated in PBS on cold bath (dry ice and methanol). PBS was then replaced with metabolite extraction buffer (MEB, 50% LC–MS grade methanol (Fisher Scientific), 30% LC–MS grade acetonitrile (Fisher Scientific) and 20% ultrapure water), 1 ml extraction buffer for $1 \times 10^6$ cells; after a couple of minutes on the cold bath, cells were stored at −80 °C overnight. The next day, extracts were scraped and mixed agitating for 15 min at 4 °C at maximum speed (3000 rpm). Finally, extracts were centrifuged for 10 min at maximum speed (14,000 rpm) at 4 °C and transferred into LC–MS vials for analysis. Valine-d8 5 µM (CK isotopes) was used as internal standard for the MEB.

For the ¹³C-glucose labelling experiments with the FOXO1 inhibitor (AS1842856) (Merck), SW48 isogenic cells were plated on 6-well plates at a density of $8 \times 10^5$ cells/well in full media conditions (5 replicates for each cell line). 24 h later cells were pre-treated with DMSO (0.003% v/v) control or FOXO1 inhibitor (1 µM) for 12 h in full media conditions. Cells were then washed with PBS and media replaced with low nutrient media where 2 mM glucose was substituted with equimolar concentration of ¹³C-labelled glucose (D-glucose, CK Isotopes) and DMSO (0.003% v/v) control and FOXO1 inhibitor (1 µM) were replenished for further 12 h until extraction.

To assess the immediate metabolic fluxes, we used 4-h tracing in all nitrogen tracing experiments. For nitrogen labelling experiments, SW48 isogenic cells were plated on 6-well plates at a density

of $8 \times 10^5$ cells/well in full media conditions (5 replicates for each cell line). 24 h later, cells were washed once with PBS and cultured in low nutrient media where 2 mM glutamine was substituted with equimolar concentrations of either L-glutamine amide-$^{15}$N (CK Isotopes) or L-glutamine α-$^{15}$N (CK Isotopes) for 4 h before extraction.

For the ammonia labelling experiments, SW48 isogenic cells were plated on 6-well plates at a density of $8 \times 10^5$ cells/well in full media conditions (5 replicates for each cell line). 24 h later, cells were washed once with PBS and cultured in low nutrient media containing 3 mM of $^{15}$N-labelled ammonium chloride (CK Isotopes) for 4 h before extraction. We repeated this experiment where 24 h after seeding, cells were pre-treated with DMSO (0.003% v/v) control or FOXO1 inhibitor (1 μM) for 12 h in full media conditions. Cells were then washed with PBS and media replaced with low nutrient media containing labelled ammonia with either DMSO control (0.003% v/v) or FOXO1 inhibitor (1 μM) for further 4 h before extraction.

## Cell treatments for SRB assays

LIM1215 and SW48 cells were seeded at $3 \times 10^3$ cells/well and $7 \times 10^3$ cells/well densities, respectively. 24 h after seeding and complete attachment, media was replaced with either full media or low nutrient media containing the single drug or combined drug treatments below and cultured for an additional 72 h.

For glutamine treatments, serial two-fold dilution of glutamine (maximum concentration 5 mM) was performed in full media or low nutrient media.

For drug treatments in low nutrient conditions, serial two-fold dilution of GLS inhibitor (CB839, maximum concentration of 100 μM, and DMSO control of maximum 0.05% v/v) or GLUL inhibitor (MSO, maximum concentration of 100 mM dissolved in low nutrient media) was performed.

For drug combination experiments, serial dilution of CB839 (Selleckchem) or MSO (Sigma) at concentrations above was performed in the presence or absence of sublethal doses of MSO (2 mM) or CB839 (100 nM), respectively. Water (1% v/v for MSO) or DMSO (0.05% v/v for CB839) were used as vehicle controls.

For single-dose drug treatments, we used the FOXO1 inhibitor (AS1842856) at 100 nM, MSO at 2 mM, CB839 at 100 nM, and KRAS G12D inhibitor (MRTX1133) at 100 nM. Water (1% v/v for MSO), DMSO (0.05% v/v for CB839, 0.003% v/v for AS1842856 and 0.002% for MRTX1133) were used as vehicle control.

The BLISS assay was also performed with the same protocol using 96-well plates using a two-fold dilution series of MSO (highest concentration of 25 mM) and CB839 (highest concentration 12.5 μM) in an $8 \times 8$ checkerboard pattern of combinations (Lin et al, 2012). The BLISS assay analysis was performed with the online Synergy Finder tool (www.synergyfinder.org).

At the end of the 72 h growth assay, all cells were fixed and subjected to the SRB assay described in the methods. Cell viability is represented as the percentage survival normalised to all controls indicated above.

## Western blots

SW48 cells were seeded at $5 \times 10^5$/well on a 6-well dish in full media. 24 h later, the media was changed to either full media or

low-nutrient media for further 48 h. For MRTX1133 treatment, 24 h after seeding, cells were treated either with MRTX113 (100 nM) or DMSO control (0.002% v/v) in full or low-nutrient conditions for further 48 h. Cells were then extracted from 6-well plates using 80 μl/well RIPA buffer (300 mM NaCl, 1.0% NP-40, 0.5% sodium deoxycholate, 0.1% SDS (sodium dodecyl sulphate), 50 mM Tris-HCl, pH 8.0 with protease and phosphatase inhibitor 2/3 cocktails). Lysates were incubated on ice, agitating every 10 min for 30 min. Lysates were then centrifuged for 10 min at 14,000 rpm at 4 °C and the supernatant processed for protein quantification using Pierce BCA protein colorimetric assay (ThermoFisher Scientific) following the manufacturer's instructions. Absorbance was measured with a TECAN spectrophotometer at 562 nm. Proteins were diluted in 1x NuPAGE LDS Sample Buffer with final of 80 mM DTT (Thermo Scientific) and heated at 95 °C for 5 min. Samples were run on 4-12% NuPAGE Bis-Tris gels (Thermo Scientific) at constant 120 V for 1 h in 1x NuPAGE MOPS SDS Running Buffer (Thermo Scientific). Wet transfer of the proteins to a PVDF (polyvinylidene difluoride) membrane (Immobilon-P, Merck) was done using BioRad transfer system (Mini Trans-Blot® Cell, Bio-Rad). Membranes were then stained with Ponceau to assess protein transfer. Afterwards, the membranes were blocked for 1 h with blocking buffer (5% BSA or 5% milk in TBST (Tris-Buffered Saline and 0.1% Tween). Primary antibody incubation was done overnight at 4 °C.

Primary antibodies used are FOXO1 monoclonal antibody (ThermoFisher Scientific, 1:1000), AKAP12 (Sigma, 1:1000), glutamine synthase (Abcam, 1:1000), anti-RAS antibody (Merck, 1:1000), phospho-4E-BP1 Thr37/46 (Cell Signalling, 1:1000), phospho-p44/42 MAPK (ERK1/2; Cell Signalling, 1:1000), p44/42 MAPK (ERK1/2; Cell Signalling, 1:1000), phospho-Akt Ser473 (clone D9E; Cell Signalling, 1:1000), Akt (pan, clone 40D4; Cell Signalling, 1:1000), anti-ß actin (Merck, 1:1000), histone H3 (Cell Signalling, 1:1000), alpha-tubulin (Santa Cruz, 1:1000), anti-HSP90 (BD Biosciences, 1:10,000).

The next day, membranes were washed 4 times in TBST and incubated with LI-COR secondary antibodies (LI-COR, IRDye-680 or IRDye-800; anti-mouse and anti-rabbit) diluted 1:10,000 in blocking buffer for 1 h at room temperature. After 4 washes with TBST, membranes were imaged using LI-COR Odyssey CLx scanner. Western Blots were quantified with Image Studio.

## Active RAS pull-down assay

This assay was performed using Active RAS Pull-Down and Detection Kit (ThermoFisher Scientific) according to the manufacturer's instructions. For this assay, $1 \times 10^7$ SW48 cells were seeded on 10 cm dishes. After 24 h, cells were washed with 1xPBS and harvested using 500 μl/dish 1X Lysis/Binding/Wash Buffer (provided in the kit). Lysates were kept on ice, agitating every 10 min for 30 min. Lysates were then centrifuged at 14,000 rpm at 4 °C, and the supernatant was processed for protein quantification using Pierce BCA colorimetric assay (ThermoFisher Scientific) following the manufacturer's instructions. 100 μl of Glutathione resins (provided in the kit) were used per sample, the supernatant was discarded, and 400 μl of 1X Lysis/Binding/Wash Buffer with 80 μg of GST-RAF1-RBD was added to the beads in each tube. An additional 700 μl of 1X Lysis/Binding/Wash Buffer with a total of 1.5 mg protein was added to the beads in each tube and incubated

overnight at 4 °C with gentle rotation. The next day, samples were washed 4 times with 1X Lysis/Binding/Wash Buffer using the columns provided in the kit. The protein was then eluted from the column with 50 µl of reducing sample buffer (2.5 µl ß-mercaptoethanol in 50 µl 2x SDS buffer (provided by the kit)). Eluted samples were then heated for 5 min at 95 °C and run on 4–12% Bis-Tris gel (ThermoFisher Scientific) as described earlier. Samples were then blotted for total Ras (Merck) and Actin (Merck).

## Nuclear and cytoplasmic fractionation assay

This assay was performed using NE-PERNuclear and Cytoplasmic Extraction kit (ThermoFisher Scientific). Briefly, SW48 and LIM1215 cells were seeded on a 6-well plate ($1 \times 10^6$ cells/well and $8 \times 10^5$). 24 h later, cells were washed with 1x PBS and collected by scraping with 80 µl/well CER-I buffer containing protease (Merck) and phosphatase (Merck) inhibitors. Cells were vortexed for 15 s and placed on ice for 15 min. 4.4 µl/sample CER-II buffer was added, vortexed for 5 s and placed on ice for 2 min. After another vortex step, samples were centrifuged for 5 min on a benchtop centrifuge at maximum speed (14,000 rpm) at 4 °C. The supernatant (the cytosolic fraction) was transferred into a clean tube, and the pellet (the nuclear fraction) was resuspended in 40 µl of ice-cold RIPA buffer described earlier. Resuspended nuclear fractions were kept on ice for 1 h during which samples were vortexed for 15 s every 10 min. Samples were then centrifuged at maximum speed (14,000 rpm) for 10 min at 4 °C on a benchtop centrifuge. Samples were diluted in 1x NuPAGE LDS Sample Buffer with a final of 80 mM DTT (ThermoFisher Scientific), heated at 95 °C for 5 min, and processed further for western blotting.

## Immunostaining

SW48 ($4 \times 10^4$ cells/well) or LIM1215 ($2 \times 10^4$ cells/well) cells were seeded on 8-well removable LabTek chambers (ThermoFisher Scientific). 24 h later, cells were washed with PBS, and the media was changed to low-nutrient media. After another 24 h, cells were fixed with 4% PFA (ThermoFisher Scientific) for 15 min at room temperature, washed with PBS and blocked with blocking buffer (1% BSA, 0.1% Triton-X, 5% Goat serum in PBS) for 30 min. Cells were then stained with FOXO1 monoclonal antibody (ThermoFisher Scientific, 1:200) diluted in blocking buffer overnight at 4 °C. The next day, cells were washed with PBS and incubated with a secondary antibody (AlexaFluor 568 conjugated goat anti-rabbit IgG (ThermoFisher Scientific, 1:500) and Hoechst 33342 (ThermoFisher Scientific, 1:1000) in blocking buffer for 1 h at room temperature. Cells were washed 3 times with PBS, wells were removed, and the sample was mounted with Mowiol (5% w/v Mowiol 4-88, 12% w/v glycerol in 2:3 dilution of 0.2 M Tris buffer, pH 8.5, in water). Cells were imaged using 63X objective on Leica SP5 Confocal (SP5, Leica Microsystems). Nuclear quantification of FOXO1 was done using Cell Profiler software.

## RNA extraction and real-time qPCR

For treatment with FOXO1 inhibitor, SW48 isogenic cells were plated on 6-well dishes $8 \times 10^5$ cells/well in full media conditions. 24 h after seeding, cells were treated with either DMSO (0.003% v/v) vehicle control or FOXO1 inhibitor (1 µM) in full media conditions. After 24 h of treatment, RNA was extracted.

For treatment with MRTX1133, KRAS G12D inhibitor, SW48 isogenic cells were plated on 6-well dishes $5 \times 10^5$ cells/well in full media conditions. 24 h after seeding, cells were treated with either DMSO (0.002% v/v) vehicle control or MRTX1133 (100 nM) in low-nutrient media conditions for further 48 h before RNA extraction.

RNA was extracted using RNeasy kit (Qiagen) following manufacturer's instructions. 1 µg RNA was reverse-transcribed using SuperScript III Reverse Transcriptase kit (ThermoFisher Scientific). qPCR was performed using SYBR Green PCR Master Mix (ThermoFisher Scientific), and following primers Hs_ACTB_2_SG (Qiagen), Hs_GLUL_1_SG (Qiagen), Hs_FOXO1_1_SG (Qiagen). Data analysis was done using StepOne software.

## siRNA

Lipofectamine RNAiMAX Transfection Reagent (ThermoFisher Scientific) via reverse transfection was used for knockdown experiments, following the manufacturer's protocol. Briefly, master mix solution of Lipofectamine RNAiMAX, Opti-MEM (Thermo Scientific) and final 1 pmol/well concentration of AllStars negative control siRNA (Qiagen), Hs_GLS_6 (Qiagen), Hs_GLUL_9 (Qiagen) siRNAs were prepared and after 15 min of incubation were aliquoted into a 96-well dish. SW48$^{+/+}$, SW48$^{+/G12D}$ and SW48$^{+/G12V}$ cells were trypsinised, centrifuged and seeded in full media at a density of $7 \times 10^3$ cells/well within the wells containing the siRNA mix. The next day, media was changed to low nutrient media and cells were kept for 72 h in culture and viability was assessed using SRB assay as described above.

For FOXO1 knockdown, master mix solution of Lipofectamine RNAiMAX, Opti-MEM (Thermo Scientific) and final 25 pmol/well concentration of AllStars negative control siRNA (Qiagen), Hs_FOXO1_Flexitube (Qiagen) siRNAs were prepared and after 15 min of incubation were aliquoted into a 6-well dish. SW48 cells were trypsinised, centrifuged and seeded in full nutrient media at a density of $5 \times 10^5$ cells/well within the wells containing the siRNA mix. 24 h later, media was changed to low nutrient media and cells were further incubated for 48 h and collected for western blotting.

## Ammonia assay

SW48$^{+/+}$, SW48$^{+/G12D}$ and SW48$^{+/G12V}$ cells were seeded on a 6-well dish ($5 \times 10^5$ cells/well) in full media conditions. 24 h later, the media was changed to low nutrient conditions for another 24 h. Cells were then washed with PBS and lysed in 100 µl of ammonia assay buffer provided with the Ammonia assay kit (Abcam). Standards and samples were prepared together with the reaction mix and incubated for 1 h. Absorbances were measured with a TECAN spectrophotometer at 570 nm. Ammonia production was calculated using the protocol provided by the manufacturer.

## RNA sequencing

For all RNA sequencing studies, SW48 isogenic cells were thawed and cultured for 3 passages before sample preparation. To control for culturing batch variation, 5 replicates for each cell line were

prepared by seeding cells into 5 flasks, cells thawed from 3 independently cryopreserved vials stocked at different passages.

Briefly, 1.5 million cells were seeded on a 6 cm dish in full media conditions. 48 h later, cells were washed once with cold PBS and collected by scraping with 1 ml PBS into 1.5 ml Eppendorf tubes. Samples were centrifuged for 3 min at 1000 rpm and 4 °C. Pellets were stored at −80 °C until shipment. RNA extraction, library preparation and sequencing (20 million reads per sample) was performed by BGI Genomics (BGI Hong Kong Company Limited).

For samples treated with FOXO1 inhibitor, SW48$^{+/+}$, SW48$^{+/G12D}$ and SW48$^{+/G12V}$ cells were thawed and cultured for 3 passages before sample preparation. Cells were seeded at 1.5 million cells/well on 6 cm dishes in full media conditions. 24 h later, media was replaced with low nutrient media containing DMSO (0.003% v/v) vehicle control, FOXO1 inhibitor (1 μM) or just media control. After an additional 24 h in low nutrient conditions, cells were scraped in 400 μl/well TRIzol (ThermoFisher Scientific) and stored at −80 °C until shipment. RNA extraction, library preparation and sequencing (20 million reads per sample) was performed by BGI Genomics (BGI Hong Kong Company Limited).

### RNA sequencing data analysis

Reads were mapped to the human reference genome GRCh38 with the STAR (version 2.6.0c) splice-aware aligner (Dobin et al, 2013) using annotation from Gencode (releases 24-35). Low-quality reads (mapping quality <20) as well as known adaptor and artifact contaminations were filtered out using Cutadapt (version 1.10.0). Read counting was performed using Bioconductor package Rsubread v1.28.1 (Liao et al, 2019) and differential expression analysis with DESeq2 v1.18.1 (Love et al, 2014).

RNA sequencing data were visualised with custom Matlab (Mathworks) scripts, which are available in figure data sources. The thresholds for log2 fold changes and false discovery rates are reported in the caption of each figure.

### Metabolomics data analysis

The chromatograms generated by LC-MS were reviewed, and the peak area was integrated using the Thermo Fisher software Tracefinder 5.0. The peak area for each detected metabolite was normalised against the total ion count (TIC) of that sample to correct any variations introduced from sample handling through instrument analysis. The normalised areas were used as variables for further statistical data analysis. For $^{13}$C- and $^{15}$N-tracing analysis, the theoretical masses of $^{13}$C and $^{15}$N isotopes were calculated and added to a library of predicted isotopes. These masses were then searched with a five ppm tolerance and integrated only if the peak apex showed less than a 1% difference in retention time from the [U-$^{12}$C and U-$^{14}$N] monoisotopic mass in the same chromatogram. After analysis of the raw data, natural isotope abundances were corrected using the AccuCor algorithm (Su et al, 2017). The resulting Microsoft Excel files are available in figure data sources.

### Statistical analysis

Graphs and statistical tests were done using GraphPad Prism 8-10 (GraphPad, San Diego, CA). Details of data analysis are reported in figure captions. Statistical significance was evaluated by one-way ANOVA using Dunnetts's test, or two-way ANOVA using Sidak's, Dunnett's tests for multiple comparisons. Technical and biological repeats were performed as indicated in figure legends, and results were reported as mean with standard deviation (SD) or standard error of the mean (SEM). All experiments, except for transcriptomics and metabolomics (see respective sections), were performed at least with three biological replicates.

## Data availability

The RNA-seq results and analysis software is available in the source data files and described in the Appendix. Raw RNA sequencing data is available at the Gene Expression Omnibus (Barrett et al, 2012) with accession number GSE306286 (https://www.ncbi.nlm.nih.gov/geo/query/acc.cgi?acc=GSE306286). The matabolomics results are available in source data files. Raw LC-MS data is available at Metabolomics Workbench (Sud et al, 2016) with accession number ST004144 (https://doi.org/10.21228/M8NZ6Q).

The source data of this paper are collected in the following database record: biostudies:S-SCDT-10_1038-S44319-025-00641-z.

## Peer review information

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

## Acknowledgements

AE acknowledges the financial support provided by the CRUK with a multi-disciplinary project award (OncoLive, C54674/A27487), pump-priming funds from the CRUK Cambridge Center (C9685/A25117, C9685/A28397). AE and ARV also acknowledge financial support from Medical Research Council program grants (MC_UU_12022/1 and MC_UU_12022/8). AE also acknowledges funding from the European Union for HORIZON 2022 fLIMAGING3D (101073507) and HILIGHT (101135034, backed by Innovate UK as project number 10107542). CF was funded by the MRC Core award grant MRC_MC_UU_12022/6, and is funded by the CRUK Programme Foundation award C51061/A27453, the H2020 European Research Council Consolidator Grant (ERC819920), and by the Alexander von Humboldt Foundation in the framework of the Alexander von Humboldt Professorship endowed by the Federal Ministry of Education and Research. Work in Simon Cook's laboratory was supported by Institute Strategic Programme Grants BB/J004456/1, BB/P013384/1 and BB/Y006925/1 from UKRI-BBSRC. We want to thank Dr Laura Tronci (CF group), Dr Pablo Oriol Valls and Dr Andrew Trinh (AE group) for performing experiments in relation to the study of KRAS mutant cell lines that have not been included in this work. We want to thank also Mr Dimitrios Prymidis (CF group) for his invaluable help in the submission of metabolomics data to public repositories. The authors are indebted to the many colleagues in the professional and technical services at the MRC Cancer Unit for making this research possible. Also, the principal investigators express their gratitude towards all members of their respective teams and support staff who endured difficulties related to the COVID lockdowns and the eventual termination of the MRC Cancer Unit.

## Author contributions

**Suzan Ber**: Conceptualization; Formal analysis; Supervision; Validation; Investigation; Visualization; Writing—original draft; Writing—review and editing. **Ming Yang**: Validation; Investigation. **Marco Sciacovelli**: Conceptualization; Writing—review and editing. **Shamith Samarajiwa**: Software; Formal analysis. **Khushali Patel**: Investigation. **Efterpi Nikitopoulou**: Investigation. **Annie Howitt**: Investigation. **Simon J Cook**: Writing—review and editing. **Ashok R Venkitaraman**: Conceptualization; Supervision; Funding acquisition; Writing—review and editing. **Christian Frezza**: Conceptualization; Supervision; Funding acquisition; Writing—review and editing. **Alessandro Esposito**: Conceptualization; Data curation; Software; Formal analysis; Funding acquisition; Validation; Investigation; Visualization; Writing—original draft; Project administration; Writing—review and editing.

Source data underlying figure panels in this paper may have individual authorship assigned. Where available, figure panel/source data authorship is listed in the following database record: biostudies:S-SCDT-10_1038-S44319-025-00641-z.

## Disclosure and competing interests statement

ARV serves on the scientific advisory board of Chugai Pharmaceuticals (Japan), and is a co-founder of PhoreMost (UK) and Sentinel Oncology (UK).

