## [Peer Review File · EMBO Reports]

FOXO1 links KRAS G12D and G12V alleles to glutamine and nitrogen metabolism in colorectal cancer

Suzan Ber, Ming Yang, Marco Sciacovelli, Shamith Samarajiwa, Khushali Patel, Efterpi Nikitopoulou, Annie Howitt, Simon J. Cook, Ashok Venkitaraman, Christian Frezza, and Alessandro Esposito

Corresponding author(s): *Alessandro Esposito (alessandro.esposito@brunel.ac.uk)*

Review Timeline:

Submission Date:	18th Jun 25
Editorial Decision:	23rd Jun 25
Revision Received:	20th Aug 25
Editorial Decision:	14th Oct 25
Revision Received:	27th Oct 25
Accepted:	3rd Nov 25

Editor: Achim Breiling

Transaction Report:

Dear Dr. Esposito,

Thank you for transferring your manuscript to EMBO reports. I now went again through the manuscript and the referee reports from The EMBO Journal.

As you know, the referees think that these findings are of interest. However, they have several comments, concerns, and suggestions, indicating that a major revision of the manuscript is necessary to allow publication of the study in EMBO reports.

EMBO reports emphasizes novel functional over detailed mechanistic insight and therefore will not require that points regarding more refined mechanistic details are addressed (e.g. point 3 of referee #3). However, all concerns regarding the major conclusions of the study and all technical and experimental limitations mentioned and points regarding data presentation need to be addressed in a revised manuscript. In particular the major issues of referee #1, point 1 of referee #2 and points 1 and 2 of referee #3 need to be addressed experimentally. As the reports are below, I will not detail them here

Revised manuscripts should be submitted within three months of a request for revision. Please contact me to discuss the revision (also by video chat) if you have further questions or comments regarding the revision, or should you need additional time.

Acceptance of your manuscript will depend on a positive outcome of another round of review at EMBO reports, using the same referees.

1) a .docx formatted version of the final manuscript text (including legends for main figures, EV figures and tables), but without the figures included. Please make sure that changes are highlighted to be clearly visible. Figure legends should be compiled at the end of the manuscript text.

2) individual production quality figure files as .eps, .tif, .jpg (one file per figure), of main figures and EV figures. Please upload these as separate, individual files upon re-submission. Please make sure that all figure panels are called out separately and sequentially in the manuscript text

For more details please refer to our guide to authors:

See also our guide for figure preparation:

Moreover, please consult our guidelines for figure legend preparation:

4) a complete author checklist, which you can download from our author guidelines (<https://www.embopress.org/page/journal/14693178/authorguide>). Please insert page numbers in the checklist to indicate where the requested information can be found in the manuscript. The completed author checklist will also be part of the RPF.

Please also follow our guidelines for the use of living organisms, and the respective reporting guidelines:
<http://www.embopress.org/page/journal/14693178/authorguide#livingorganisms>

5) that primary datasets produced in this study (e.g. RNA-seq, ChIP-seq and array data) are deposited in an appropriate public database. This is now mandatory (like the COI statement). If no primary datasets have been deposited in any database, please state this in this section (e.g. 'No primary datasets have been generated and deposited').

The accession numbers and database should be listed in a formal "Data Availability" section (placed after Materials & Methods) that follows the model below. Please note that the Data Availability Section is restricted to new primary data that are part of this study.

Data availability

6) We now request the publication of original source data with the aim of making primary data more accessible and transparent to the reader. You will receive a separate email with instructions for providing source data with your revised manuscript, including information how to upload and organize the files.

8) Regarding data quantification and statistics, please make sure that the number "n" for how many independent experiments were performed, their nature (biological versus technical replicates), the bars and error bars (e.g. SEM, SD) and the test used to calculate p-values is indicated in the respective figure legends (also for EV and Appendix figures). Please also check that all the p-values are explained in the legend, and that these fit to those shown in the figure. Please provide statistical testing where applicable. Please avoid the phrase 'independent experiment', but clearly state if these were biological or technical replicates. Please also indicate (e.g. with n.s.) if testing was performed, but the differences are not significant. In case n=2, please show the data as separate datapoints without error bars and statistics. See also:

<http://www.embopress.org/page/journal/14693178/authorguide#statisticalanalysis>

9) Please add scale bars of similar style and thickness to microscopic images, using clearly visible black or white bars (depending on the background). Please place these in the lower right corner of the images themselves. Please do not write on or near the bars in the image but define the size in the respective figure legend.

10) Please also note our reference format:

12) We now use CRediT to specify the contributions of each author in the journal submission system. CRediT replaces the author contribution section. Please use the free text box to provide more detailed descriptions and do NOT provide your final

manuscript text file with an author contributions section. See also our guide to authors:
<https://www.embopress.org/page/journal/14693178/authorguide#authorshipguidelines>

13) All Materials and Methods need to be described in the main text using our 'Structured Methods' format, which is required for all research articles. According to this format, the Methods section should include a Reagents and Tools Table (listing key reagents, experimental models, software, and relevant equipment and including their sources and relevant identifiers), uploaded as separate file, and a Methods section in which we encourage the authors to describe their methods using a step-by-step protocol format with bullet points, to facilitate the adoption of the methodologies across labs. More information on how to adhere to this format as well as downloadable templates (.doc) for the Reagents and Tools Table can be found in our author guidelines (section 'Structured Methods'):

14) Please add up to 5 keywords to the manuscript and order the sections like this, using these names:
Title page - Abstract (not more than 175 words) - Keywords - Introduction - Results - Discussion - Methods - Data availability section - Acknowledgements (please include here also the funding information) - Disclosure and Competing Interests Statement - References - Figure legends - Expanded View Figure legends

15) Please make sure that all the funding information is also entered into the online submission system and that it is complete and similar to the one in the acknowledgement section of the manuscript text file.

I look forward to seeing a revised form of your manuscript when it is ready. Please use this link to submit your revision:
<https://embor.msubmit.net/cgi-bin/main.plex>

Please let me know if you have questions or comments regarding the revision.

Kind regards,

Achim

Referee #1:

Ber et al. characterized a KRAS mutant (G12) isogenic SW48 cell line panel to study possible differences in functional/phenotypic outcome of these specific mutations. They conclude that G12D and G12V mutations most significantly link to changing glutamine/nitrogen metabolism through regulation of FOXO1.

From a general (clinical) perspective it is of interest to see whether all KRAS mutations are in the end equal in terms of mediating tumorigenesis or that specific mutations show distinct features to an extend this may eventually guide mutation specific treatment. The availability of small molecules targeting G12C specifically lends credit to the idea of mutation specific treatment, albeit that this is more due to the biochemical nature rather than the specific biology that come along with this mutation.

Major issues.

As far as I can see for each mutation one cell line is being analyzed. If this is correct one wonders what the variability in outcome is when testing 3 biological replicates so three independently established cell lines with G12V; G12D etc. This multiplies the amount of work I realize this, but on the other hand I have no clue as to how much of the sometimes small difference between cell lines is due to clonal effects or not.

Same holds for the metabolomic analysis as in the figure legends it is indicated that these concern 5 technical replicates. This is nice but one really wants to see biological replicates.

With respect to FOXO1 as mediating the effect on nitrogen metabolism it should be considered that FOXO expression and neither nuclear localization is decisive predictor of activity. FOXOs are primarily regulated through post translational modifications so at present it is only suggested that FOXOs are activated by G12V and G12D but mechanistically it remains unclear, also because the presented data on AKT the major regulator of FOXO activity is not very different between the cell lines in the sense that this would explain the specific control of FOXO activity by G12V and G12D and not the other mutants. I appreciate (and agree with) the authors argument that cells should be tested in culture media that better reflect the actual metabolic environment. However, be this the case as far as I know glutamine levels/concentration between interstitial fluid and plasma do not differ that much so glutamine deprivation appears unlikely under physiological conditions (see e.g. Doglioni et al. PMID:39743589). Interstitial measurements indeed show substantial glucose deprivation but the condition tested here (2mM of glucose) is still well above the Km of the GLUT1 transporter so although reduced compared to the 25mM that many use it is still more than sufficient and not limiting.

To me the rationale (if any) why (to solve what metabolic limitation) the G12D and V cell lines start to make glutamine out of glucose is not explained or clear. The use of GS/GLS inhibitors is also not make things easier to understand, because only a combo is really effective, but completely inhibiting glutamine metabolism is probably toxic to almost every cancer cell line irrespective of their mutational status.

Hidden in supplementary S6 is validation of data on a different cell line (LIM1215) however here FOXO1 expression is increased in G12C and not G12V so in contrast to the SW48 panel and also nuclear staining is visible in all cell lines G12A and G12C as well. So to me argues that at least part of the observations are prone to biological variation between cell lines harboring the same mutation and iterates the suggestion to test more biological replicates.

Taken together interesting study but to me it is unclear how a specific KRAS mutation links to regulation of FOXO1 and how this then translates to a mutation specific (metabolic) phenotype. The data suggest to me that the link is quantitative rather than qualitative, so e.g. a two fold increase in G12C would make this mutant comparable to G12V. The part on regulation of metabolism is at present too descriptive and also does not provide mechanistic insight as to how these observations link to the specific KRAS mutations and not to the other ones, so the conclusion stated in the abstract "KRAS mutations at codon 12 induce allele specific metabolic programs" is to me not supported by the data in the sense that the authors do not show that the other alleles are not inducing these metabolic programs if expressed at e.g. a slightly higher level.

Referee #2:

In the present manuscript, the authors reveal how specific KRAS G12 mutations influence cellular metabolism in colorectal cancer. Using isogenic cell line models combined with transcriptomic, metabolomics and biochemical approaches, they demonstrate that different KRAS alleles induce distinct metabolic programs affecting glutamine metabolism and nitrogen recycling. The study highlights that FOXO1 is upregulated in KRAS G12D and G12V mutant SW48 cells, enhancing cell viability under low-nutrient conditions. Furthermore, the authors showed that simultaneous inhibition of GS and GLS can significantly reduce cell viability of certain KRAS mutant cells.

The manuscript has some interesting and potentially important findings, but there are a number of key issues that need to be addressed.

Major:

1) While the use of an isogenic colorectal cancer cell line harboring heterozygous mutations in KRAS provides a valuable and controlled system to study the functional effects of specific KRAS mutations, the exclusive reliance on these models limits the translational value of the conclusions. Such panels may not fully reflect the biology of the targeted cancer pathways. Even within the manuscript, the authors show that KRAS mutations in the SW48 panel do not affect downstream pathways, and experiments using the LIM1215 line did not fully reproduce the results. It should be noted that neither the SW48 nor the LIM1215 cells are "normally" mutant for KRAS. These mutations were introduced artificially, and it isn't clear if the cells even rely on KRAS to maintain their signaling structure.

To strengthen the impact and biological relevance of the study, it is recommended to repeat key experiments in one or more additional cell lines that harbor endogenous KRAS mutations.

2) There is a lack of statistical analysis for western blots throughout the manuscript. It is recommended to quantify band intensity and normalize to a loading control.

3) The rationale behind the choice of full media or low-nutrient media conditions is unclear in some cases. It would be helpful to explain why each condition was chosen. For instance, the authors show modest upregulation of the ERK pathway in low nutrient conditions for G12D-mutant cells, yet they study the effects of MRTX1133 under full media conditions.

4) In the first paragraph of the results in chapter three, the authors state: "Intracellular metabolomics showed that TCA (tricarboxylic acid) cycle metabolites like α -ketoglutarate and succinate are significantly lower in G12D and G12V mutant cells

(Figure 2A)." However, Figure 2A does not show a significant difference in α -ketoglutarate levels between WT and G12D cells.

5) Can the authors explain the significant decrease in viability of G12A/C and especially WT cells when glutamine concentration is increased (Figure 2C)?

6) The authors used MRTX1133 to test the dependency of FOXO1 expression on oncogenic KRAS, focusing only on G12D cells. It would be worthwhile to study how FOXO1 expression changes in response to KRAS inhibition or knockout in other KRAS-mutant cells.

7) "FOXO1 inhibition reduced the viability of all tested cell lines, particularly cells with lower nuclear FOXO1, including KRAS wild-type cells." Did the authors study how inhibition of oncogenic KRAS and FOXO1 together affects the viability of KRAS-mutant cells?

8) "While we observed a higher sensitivity of KRAS mutant cells relative to wild-type SW48 cells, the effects were only significant at high MSO concentrations (Figure 5A, left panel)". However, it is difficult to distinguish any difference in sensitivity from the curves, the error bar in the WT group appears quite high.

9) The G12A cells show much higher sensitivity to the combination of GLUL and GLS inhibitors than G12D/V cells, especially under low nutrient conditions. This suggests that targeting these two enzymes may be more promising in treating tumors with KRAS G12A mutations than focusing on G12D/V-mutant cells.

10) "Knocking down the two genes by siRNA also confirmed the sensitivity of G12D and G12V mutant cells to the inhibition of GLS and GS together (Figure S5G)." The authors probably meant Figure S5C. Since G12A cells respond better to the inhibition of GLUL and GLS, it would be worthwhile to study how knockdown of these genes affects the viability of G12A cells.

11) "To mitigate the risk that the observed sensitivities entirely depend on the SW48 cell line background, we repeated crucial experiments using a second isogenic panel derived from the colorectal cancer cell line LIM1215 (Figure S6). The LIM1215 panel harboured the same mutant alleles, also exhibited upregulation of FOXO1, and revealed the same sensitivity of mutant LIM1215 to combinatorial drugging of glutamine synthetase and glutaminase, as observed in SW48 isogenic panel."

It is recommended to expand this part of the manuscript. Due to the poor quality of the western blots, it is difficult to draw conclusions about FOXO1 levels between WT and mutant cells. Also, Bliss analysis patterns differ between the SW48 and LIM1215 lines.

Minor:

I suggest the authors consider expanding the introduction by focusing on metabolic pathways that were studied in the manuscript.

Referee #3:

The study by Ber et al reports the results of a comprehensive transcriptomic and metabolomic characterisation of an isogenic cell line panel derived from SW48 colorectal cancer (CRC) cells harbouring different mutants of the Ras oncogene (V12G, V12D, V12C and V12A compared to WT). The authors initially perform RNA sequencing across the cell line panel to identify deregulated gene signatures linked to Ras mutations with high prevalence in CRC (V/D compared to G/C). This revealed that FOXO1 signalling is induced in D/V mutants. The authors also perform metabolomic characterisation using ¹³C-glucose tracing to show altered glutamine metabolism. They expand the analysis of FOXO1 regulation by showing enhanced nuclear localisation in D/V cells. This was most prominent in G12D and G12V mutants. The authors next implicate FOXO1 in the regulation of glutamine metabolism using a FOXO1 inhibitor. The authors then move on to using ¹⁵N-glutamine tracing to reveal that Ras regulates ammonia recycling via FOXO1-dependent regulation of GS. Finally, they show that combined inhibition of GS and GLS limits survival of Ras mutant cancer cells and this is enhanced by low nutrient conditions. The authors conclude that targeting ammonia recycling could be a therapeutic target in Ras mutant cancers.

While the manuscript contains some interesting findings, its overall quality is quite difficult to assess. The data are presented in a highly confusing manner, with the figures not being discussed in the correct order and jumping from main to supplementary data without clear reason. While most of the experiments were done in all cell lines, the investigation of altered ammonia metabolism focusses only on the G12D and G12V mutants, making it difficult to connect to the inhibitor results. The manuscript contains extensive supplementary material, some of it being unrelated to the main line of investigation. The discussion either reiterates the results section or overstates the findings and requires substantial editing for clarity. In addition, page and line numbers are missing making it hard to conduct the review. Despite these major limitations, the manuscript reports some interesting findings, most importantly the somewhat unexpected observation that mutant Ras activates FOXO1. However, the authors need to address the inconsistencies in the data to provide sufficient support for their conclusion before the work can be published.

Major points:

- 1) The initial cell line characterisation is confusing. Why do the Ras mutant cells not show activation of ERK and AKT signalling? The experiment shown in S1B and C should have been performed after growth factor deprivation. S1D uses 1% FCS but also reduced glucose (2 mM), which could affect the activity of these signalling pathways. As all other experiments built on these Ras mutations, it is mandatory that their functionality is proven.
- 2) A major limitation of the study is that the experiments switch between full media and low nutrient conditions. The exact formulation of the low nutrient medium (2 mM glucose, 1% FCS) is quite arbitrary, despite the claims made in the discussion that this may represent the in vivo condition and combines glucose withdrawal with reduced availability of serum-derived growth factors and lipids. Nevertheless, the "low nutrient" condition reveals the more interesting phenotypes, suggesting that all experiments should have been done like this.
- 3) The mechanism by which mutant Ras induces FOXO1 activation remains unclear. While the increased nuclear localisation seems highly relevant (Fig. 3E and F), the authors also show some minor regulation of FOXO1 mRNA (Fig. S3B). A more detailed investigation of this mechanism would greatly enhance the impact of the study.
- 4) The authors claim that oncogenic Ras regulates expression of GLUL/GS via FOXO1. However, inhibition of Ras using MRTX1133 does not reduce GS expression in full media (Fig. 3C), while the induction is only seen in G12D cells in the low nutrient condition. Did the inhibitor work at all? Similarly, siRNA-mediated deletion of FOXO1 only had a minor effect on GLUL mRNA expression, and baseline GLUL expression did not follow the suggested pattern of induction by the two prevalent Ras mutations (Fig. S3E).
- 4) The observation that combined inhibition of GS and GLS reduces viability of mutant Ras cells is interesting. However, this effect seems to be strongest in G12A cells, which only show minor activation of FOXO1 and GS. How does this square up with the conclusions drawn by the authors that this could be a strategy to eliminate mutant Ras cancers?

Minor points:

- The Waddington scheme used in the graphical abstract implies that the study investigated "metabolic trajectories" induced by different Ras mutants, which is an overstatement. This should be removed.
 - Fig. S1A is difficult to understand and not relevant for the study.
- Page 7 middle paragraph should be "An increase in nuclear FOXO1...".
- The statistical analysis in Figure 5D should be checked. Only biologically independent replicates (n=4) should be used for the calculation.
 - The limitation section raises some valid points but also includes several statements that merely describe standard procedures to assure robustness of the data. Using sufficient replicates is just good scientific practice but does not support the conclusions.
 - Showing individual tissue cores copied from the Human Proteome Atlas is very arbitrary and should not be included (Fig. S7D).

KRAS drives FOXO1–GLUL axis

Dear reviewers,

we are grateful for the opportunity that Dr Breiling and Dr Klimmeck gave us to resubmit the manuscript to EMBO Reports. We would like to thank you for the thorough feedback provided at EMBO Journal that, despite the negative recommendation, offers us the opportunity to improve our manuscript further.

Given the length of our revision, we have structured our response in several sections:

- Note 1 on cell lines and reproducibility.
- Note 2 on mechanisms of FOXO1 regulation.
- Note 3 on low nutrient conditions.
- Note 4 on sensitivity of the G12A cell line.
- Point-by-point response.

We have worked to strengthen our work and, at the same time, to make the limitations of our work even more transparent than before. We hope you will appreciate our detailed revision and consider it an asset for EMBO Reports.

Best regards,

Alessandro

Note 1. Cell lines and reproducibility.

Ref 1. “As far as I can see for each mutation one cell line is being analyzed. If this is correct one wonders what the variability in outcome is when testing 3 biological replicates so three independently established cell lines with G12V; G12D etc. This multiplies the amount of work I realize this, but on the other hand I have no clue as to how much of the sometimes small difference between cell lines is due to clonal effects or not.”

Ref 2. “While the use of an isogenic colorectal cancer cell line harboring heterozygous mutations in KRAS provides a valuable and controlled system to study the functional effects of specific KRAS mutations, the exclusive reliance on these models limits the translational value of the conclusions. Such panels may not fully reflect the biology of the targeted cancer pathways. Even within the manuscript, the authors show that KRAS mutations in the SW48 panel do not affect downstream pathways, and experiments using the LIM1215 line did not fully reproduce the results. It should be noted that neither the SW48 nor the the LIM1215 cells are “normally” mutant for KRAS. These mutations were introduced artificially, and it isn't clear if the cells even rely on KRAS to maintain their signaling structure. To strengthen the impact and biological relevance of the study, it is recommended to repeat key experiments in one or more additional cell lines that harbor endogenous KRAS mutations.”

We acknowledge that using multiple, independently established isogenic clones for each mutation would be the ideal scenario to rule out clonal effects. However, such independently derived clones for these specific KRAS mutations in the SW48 and LIM1215 backgrounds are not available. To address reproducibility and the potential for clonal artefacts, we have employed several strategies and propose changes to the manuscript.

- **Focus on robust changes** (addressing the comment “I have no clue as to how much of the sometimes small difference between cell lines is due to clonal effects or not.”): Throughout our study, while noting various subtle differences, we have prioritised characterising and drawing conclusions from metabolic and transcriptional changes that are not only statistically significant but also exhibit substantial effect sizes. FOXO signalling and metabolic-related signatures were amongst the strongest emerging from RNAseq analysis (Fig. 1A-C, S1F-J, File S1) that we executed with rather stringent experimental (5 independent cultures and different passages) and statistical conditions (FDR<0.05, fold-change>2). Even at the single gene level, FOXO1 mRNA is approximately 3-fold higher in SW48 G12D cells compared to wild-type (WT), with a p-adjusted value of $\sim 10^{-58}$ (File S1).
Changes revealed by metabolomics are equally substantial both statistically and in effect size.
- **Allele-specific inhibitor control (for G12D)**: For the SW48 G12D mutant, we provide strong evidence that the observed phenotypes are directly linked to the G12D allele and not a random clonal effect. Treatment with MRTX1133, a highly specific KRAS G12D inhibitor, selectively kills the SW48G12D mutant (Fig S5E, right panel). Moreover, MRTX1133 treatment abrogates the increased FOXO1 expression (Figs. 3C-D, S3B, S3F) and reverses the sensitivity to the MSO/CB839 drug combination in SW48 G12D cells, while having no such effects on the parental WT cells (Fig. S5E). This demonstrates a direct causal link between the G12D allele and these key phenotypes in the G12D cell line.
- **Validation in a second cell line panel (LIM1215)**: We utilised a second isogenic panel derived from the LIM1215 colorectal cancer cell line. While acknowledging some allele-specific differences in the precise patterns of FOXO1 expression between the SW48 and LIM1215 backgrounds (which may reflect genuine biological context-dependency and

will be discussed – see proposed changes), the key functional vulnerability – the increased sensitivity of KRAS mutant LIM1215 cells to the MSO/CB839 combination – was conserved (Fig. S6A, B). Of particular note, the clones that exhibit higher FOXO1 protein expression, do exhibit enhanced synergy between the drugs. This provides an essential layer of validation for our therapeutic conclusions across different genetic backgrounds.

While we cannot provide independently derived clonal lines for all mutants, these approaches, particularly the MRTX1133 data for G12D, the corroborating findings in the LIM1215 panel for the therapeutic vulnerability, provide substantial confidence that our main conclusions are not driven by random clonal effects but are linked to the specific KRAS mutations under study.

Changes to the manuscript

Method section.

To clarify what we defined as a replicate, we moved information from the section describing the limitations of the study to the RNA sequencing section of the methods.

Previous text: *“SW48 isogenic cells were thawed and cultured for 3 passages before sample preparation. 5 replicates from 3 different batches of each cell line were used for the RNAseq experiment.”*

Proposed change: *“For all RNA sequencing studies, SW48 isogenic cells were thawed and cultured for 3 passages before sample preparation. To control for culturing batch variation, 5 replicates for each cell line were prepared by seeding cells into 5 flasks, cells thawed from 3 independently cryopreserved vials stocked at different passages.”*

Results section.

To clarify the results obtained with LIM1215 as controls for clonal artefacts, including the variabilities observed, we amended the second-to-last paragraph of the results section.

Previous text: *“The LIM1215 panel harboured the same mutant alleles, also exhibited upregulation of FOXO1, and revealed the same sensitivity of mutant LIM1215 to combinatorial drugging of glutamine synthetase and glutaminase, as observed in SW48 isogenic panel.”*

Amended text: *“The LIM1215 panel harboured the same mutant alleles and exhibited upregulation of FOXO1. Specific patterns of FOXO1 expression and nuclear localisation showed some variation between the two cell line backgrounds (e.g., Figure S6C,D compared to Figure 3A,E,F), with the G12D clones exhibiting a stronger phenotype. The LIM1215 mutant cells revealed a conserved vulnerability to the combinatorial inhibition of glutamine synthetase and glutaminase similar to that observed in the SW48 isogenic panel. Most importantly, the clones that exhibit higher expression of the FOXO1 protein do exhibit enhanced synergy between the drugs. This suggests that the reliance on the glutamine-glutamate cycle might be a common feature of KRAS mutant cells in different colorectal cancer contexts.”*

To emphasise that FOXO signalling is significantly different in the G12D and G12V, we amended the text in the first paragraph of the results section.

Previous text: *“Notably, both PCA and gene enrichment analysis (Figure 1B,C) suggest an involvement of FOXO1 in the differences observed in the G12D and G12V mutants.”*

Amended text: *“Notably, both PCA and gene enrichment analysis of our transcriptomic data (Figure 1B,C) revealed a significant FOXO signaling signature associated with the G12D and G12V mutants, including the upregulation of FOXO1 itself alongside other known FOXO target genes and pathway components (see also Table S1 and File S1), indicative of an enhanced FOXO1-driven transcriptional program in these cells.”*

Limitation of the study

To address various referees' comments and changes we proposed, we have re-written this section.

“This study utilised in vitro cell culture models. While we employed 'low nutrient' conditions to better reflect aspects of the in vivo tumour microenvironment and applied rigorous cell line characterisation and authentication, these models cannot fully recapitulate the complexity of three-dimensional tissues or systemic physiology.

Our use of commercially available isogenic cell line panels means our findings are based on single clones for each KRAS mutation. While the generation of multiple, independently derived clones was beyond the scope of this study, we mitigated potential clonal effects by: i) focusing on robust phenotypes, ii) demonstrating direct KRAS G12D allele-dependency for key effects using the specific inhibitor MRTX1133, and (iii) validating the central therapeutic vulnerability in a second colorectal cancer isogenic cell line panel (LIM1215), which showed conserved sensitivity despite some inter-line variability in upstream molecular patterns.

While we establish a functional KRAS-FOXO1-Glutamine metabolism axis, particularly for G12D/V alleles, the precise upstream molecular mechanisms linking each distinct KRAS G12 variant to FOXO1 regulation (e.g., specific PTMs) were not exhaustively detailed.”

Note 2. Mechanisms of FOXO1 regulation.

Ref 1. *“With respect to FOXO1 as mediating the effect on nitrogen metabolism it should be considered that FOXO expression and neither nuclear localization is decisive predictor of activity. FOXOs are primarily regulated through post translational modifications so at present it is only suggested that FOXOs are activated by G12V and G12D but mechanistically it remains unclear, also because the presented data on AKT the major regulator of FOXO activity is not very different between the cell lines in the sense that this would explain the specific control of FOXO activity by G12V and G12D and not the other mutants.”*

Ref 3. *“The mechanism by which mutant Ras induces FOXO1 activation remains unclear. While the increased nuclear localisation seems highly relevant (Fig. 3E and F), the authors also show some minor regulation of FOXO1 mRNA (Fig. S3B). A more detailed investigation of this mechanism would greatly enhance the impact of the study.”*

We agree with the referees' comments. As guided by the Editors regarding the transfer of our manuscript to EMBO Reports, further characterisation of the upstream mechanistic details of how mutant KRAS instructs FOXO1 activity is considered beyond the scope of the revisions. Therefore, while we acknowledge the value of such a mechanistic investigation, our revisions focus on clearly presenting the observed link between KRAS mutations and

altered FOXO1 status, and thoroughly characterising the functional downstream consequences of this on glutamine/nitrogen metabolism and cellular vulnerabilities, which is the primary thrust of our manuscript. The detailed upstream mechanisms connecting specific KRAS effector pathways to FOXO1 regulation remain a vital area for future dedicated studies.

However, we provide feedback and suggest minor changes to the manuscript to clarify the potential mechanisms further.

Referee 1 correctly points out that FOXO1 activity is complex and regulated by multiple post-translational modifications beyond just its expression levels or nuclear localisation. We agree that our data on total AKT phosphorylation (pS473) in SW48 mutant cell lines did not show apparent differences that would readily explain a specific G12D/V effect on FOXO1 via this kinase alone, particularly under low-nutrient conditions (Fig. S1B, S1C). Referee 3 also correctly points out that we did not report the exact molecular mechanisms by which KRAS regulate FOXO1.

KRAS can influence FOXO1 through various pathways. We observed a modest upregulation of ERK pathway activity in G12V and G12D cells under low-nutrient conditions (Fig. S1C), and ERK is known to phosphorylate and regulate FOXO1. Furthermore, we noted a significant increase in the expression of AKAP12 in several mutant lines, particularly G12D (Fig. S1B, S1C). AKAP12 is a scaffold protein implicated in PKA signalling, which can also enhance FOXO1 transcriptional activity. While dissecting the precise upstream KRAS→FOXO1 signalling cascade and the specific PTMs involved is an extensive undertaking and, as per the editor's guidance for EMBO Reports, not a primary focus for this revision, our study provides substantial evidence for the functional relevance of altered FOXO1 in KRAS mutant cells:

Transcriptional signature: Our RNA sequencing analysis revealed a clear FOXO signalling signature in G12D and G12V mutant cells, including the upregulation of FOXO1 itself and known FOXO target genes (Fig. 1B, 1C, File S1). This indicates an enhanced FOXO1-driven transcriptional program.

Dependence on mutant KRAS (G12D): In SW48 G12D cells, treatment with the G12D-specific inhibitor MRTX1133 led to a significant reduction in FOXO1 protein expression (Fig. 3C-D, S3B, S3F). Importantly, MRTX1133 also decreased the expression of the key FOXO1 transcriptional target, glutamine synthetase (GS/GLUL), specifically under low-nutrient conditions (Fig. 3D).

Functional role of FOXO1 in metabolism: Inhibition of FOXO1 using AS1842856 (iFOXO1) downregulated GLUL mRNA expression in G12D and G12V cells (Fig. 3G) and, critically, abrogated the enhanced synthesis of glutamine from glucose that is characteristic of these KRAS mutant lines (Fig. 3H). This directly links FOXO1 to the observed metabolic reprogramming.

Therefore, although we did not delineate the exact post-translational modifications on FOXO1, our data strongly support a model in which oncogenic KRAS (exemplified by G12D) upregulates FOXO1 levels and its nuclear presence, leading to an enhanced FOXO1-dependent transcriptional program that includes GLUL. This, in turn, drives key changes in glutamine and nitrogen metabolism. We acknowledge in the manuscript that the precise mechanisms by which different KRAS alleles differentially engage pathways leading to FOXO1 modulation are an area for future detailed investigation. Still, our current findings highlight a novel and functionally crucial downstream consequence.

Changes to the manuscript

We revised the discussion of mechanisms likely to be mediators of the increased FOXO1 activity in mutant cell lines.

Previous text: *“Differences in the GTPase activity of KRAS or interaction with other proteins might be underpinning the different expression or activities of key regulators of FOXO1 such as AKAP12 (involved in PKA-mediated FOXO1 phosphorylation), AKT and ERK (Figure S1C-D) can alter expression, localisation and activity of FOXO1. Although so far not within the context of KRAS-driven carcinogenesis, AKT is known to phosphorylate FOXO1 to induce interaction with the protein 14-3-3 and to induce its translocation to the cytoplasm where FOXO1 can be degraded by the ubiquitin-proteasome system. PKA and ERK also phosphorylate FOXO1 enhancing its transcriptional activity. Interestingly, PKA is associated with AKAP12, a scaffold protein that we and others have shown to be highly expressed in response to some mutant KRAS G12 alleles (Figure S1C-D).”*

Amended text: *“These inherent allele-specific biochemical properties likely contribute to the qualitative and quantitative differences in the resulting metabolic landscapes and underpin the observed variations in FOXO1 expression, nuclear localisation, and overall functional state.*

Although not in the context of oncogenic KRAS, FOXO1 is known to be phosphorylated by AKT, ERK and PKA. We do not observe significant changes in AKT (phosphor-Ser473) activation across mutant lines, but we detect a robust increase in the phosphorylation of 4EBP1 downstream of the PI3K-AKT-mTOR axis (Figure S1BC). More importantly, G12D and G12V mutant lines exhibit higher phosphorylated ERK under low-nutrient conditions (Figure S1C) and a vast increase in the expression of the scaffold protein AKAP12 (also present in G12C to a lesser extent; Figure S1BC), a protein directly implicated in PKA-mediated phosphorylation of FOXO1. While the precise contribution of each pathway and the specific post-translational modifications on FOXO1 were beyond the scope of this study, it is plausible that KRAS-driven alterations in PI3K/AKT/mTOR, ERK and PKA signalling not only modulate FOXO1 functional state but also drive its mRNA levels through FOXO1 transcriptional autoregulation”

Note 3. Low nutrient conditions.

Ref 1. *“I appreciate (and agree with) the authors argument that cells should be tested in culture media that better reflect the actual metabolic environment. However, be this the case as far as I know glutamine levels/concentration between interstitial fluid and plasma do not differ that much so glutamine deprivation appears unlikely under physiological conditions (see e.g. Doglioni et al. PMID:39743589). Interstitial measurements indeed show substantial glucose deprivation but the condition tested here (2mM of glucose) is still well above the Km of the GLUT1 transporter so although reduced compared to the 25mM that many use it is still more than sufficient and not limiting.”*

Ref 2. *“The rationale behind the choice of full media or low-nutrient media conditions is unclear in some cases. It would be helpful to explain why each condition was chosen. For instance, the authors demonstrate modest upregulation of the ERK pathway in low-nutrient conditions for G12D-mutant cells; however, they study the effects of MRTX1133 under full media conditions.”*

The referee correctly notes that interstitial glutamine concentrations are generally substantial, and our study does not primarily model a state of glutamine deprivation. Indeed, studies such as Sullivan et al. (2019) (our ref 27) demonstrate that glutamine can be relatively abundant within the tumour microenvironment of KRAS-driven cancers, sometimes

even more so than in plasma. Our findings align with this, showing that KRAS G12D and G12V mutant cells exhibit enhanced resilience or even a growth advantage when cultured with high glutamine under conditions of reduced glucose and serum (Fig. 2C bottom panel, Fig. 2D). Our focus is therefore on their intrinsic capacity to upregulate glutamine and nitrogen metabolism, including *de novo* synthesis of glutamine from glucose (Figs. 2A,B and Fig. 3H) and ammonia recycling (Fig. 4A,B), in abundant glutamine conditions, which becomes particularly relevant under broader nutrient stress where glucose availability is reduced.

Regarding glucose, while we acknowledge that 2 mM glucose is above the typical K_m for GLUT1, this concentration represents a significant reduction from standard culture media (e.g., 11 mM RPMI). This is a deliberate step towards the lower glucose levels observed in actual tissue interstitial fluid (e.g., Nightingale et al., 2019; our ref 26) and, more specifically, within tumor microenvironments, which are often characterised by glucose concentrations substantially lower than plasma or standard media (Sullivan et al., 2019; our ref 27). Our intention in using 'low nutrient' media (2 mM glucose and 1% FCS) was not to simulate complete glucose exhaustion. Instead, the aim was to impose a significant metabolic challenge by substantially reducing a primary carbon source and concurrently diminishing growth factor support. This approach allows us to unmask intrinsic metabolic adaptations and differential vulnerabilities that are often obscured in nutrient-replete standard culture conditions.

Crucially, these specific 'low nutrient' conditions were effective in revealing distinct, allele-specific phenotypes. These observations highlight that specific KRAS alleles can drive metabolic reprogramming, conferring a context-dependent advantage. While our *in vitro* system is a model, it has allowed us to identify a novel KRAS-FOXO1-glutamine metabolism axis under conditions of applied metabolic stress, which are more aligned with aspects of the *in vivo* nutrient landscape than standard high-nutrient media. We will ensure the manuscript text clearly frames these conditions accordingly.

Changes to the manuscript

We expanded the rationale for our choice within the results section.

Previous text: *“Given that growth media used for cell culture are formulated with supraphysiological concentrations of nutrients, we repeated the SRB assay with lower concentrations of serum (1%) and glucose (2 mM). These values are more representative of tissue interstitial levels, and tumour microenvironment.”*

Amended text: *“Recognising that standard cell culture media often contain supraphysiological nutrient levels, we performed key functional assays under low nutrient conditions (2mM glucose and 1% FCS). This glucose concentration, while not modelling severe hypoglycaemia, is substantially reduced from standard RPMI and falls within the range of glucose levels reported in various tumour microenvironments (0.2- 2.5 mM), which are often considerably lower than plasma levels (~5 mM). This reduction, coupled with lowered serum, was intended to impose a relevant metabolic stress to unmask differential adaptations and vulnerabilities.”*

Note 4. Sensitivity of the G12A cell line.

Ref 2. *“The G12A cells show much higher sensitivity to the combination of GLUL and GLS inhibitors than G12D/V cells, especially under low nutrient conditions. This suggests that targeting these two*

enzymes may be more promising in treating tumors with KRAS G12A mutations than focusing on G12D/V-mutant cells.”

Ref 3. 4) [5] The observation that combined inhibition of GS and GLS reduces viability of mutant Ras cells is interesting. However, this effect seems to be strongest in G12A cells, which only show minor activation of FOXO1 and GS. How does this square up with the conclusions drawn by the authors that this could be a strategy to eliminate mutant Ras cancers?

We thank the referees for highlighting the pronounced sensitivity of the KRAS G12A mutant.

Our primary mechanistic focus was on G12D/V due to their clinical prevalence and clear FOXO1-GLUL engagement. G12A is a rare mutation, resulting in its standalone therapeutic targeting of a lesser immediate clinical breadth, although its biology is instructive.

The heightened sensitivity of SW48 G12A cells to combined GS/GLS inhibition (**Figure 5**), despite the lack of FOXO1/GS activation compared to G12D/V (**Figure 2E,F, 3A,F**) might seem surprising at first instance. We propose that this distinct sensitivity arises because:

1. G12A cells exhibit lower basal GLUL/GS expression (**Figure 2E,F**) and consequently, minimal *de novo* glutamine synthesis from glucose (**Figure 2A,B**).
2. Unlike G12D/V cells, which can be net glutamine producers, G12A cells are net consumers of exogenous glutamine (**Figure 2B**, top panel).
3. However, the G12A mutation drives a more active, glycolytic (**Figure 1D**), metabolic state than wild-type cells, with higher carbon flux through the TCA cycle (**Figure S2F**) and increased demands for biosynthesis (**Figure S2D**).

Thus, inhibiting both GLS and GS with their minimal activity is therefore highly detrimental. Wild-type cells, despite also having low GS, likely have lower overall glutamine demand under stress and/or greater metabolic flexibility (**Figure 5D**).

This finding broadens the potential of targeting the glutamine-glutamate cycle. While the reasons for dependency may differ (FOXO1-GLUL upregulation in G12D/V vs low GLS, GS/high demand in G12A), the vulnerability itself is shared among several KRAS mutant types. We therefore consider this a positive outcome, suggesting that therapies disrupting this cycle could be relevant for a broader range of KRAS-mutant tumours than initially hypothesised based on the G12D/V-FOXO1 axis alone.

Changes to the manuscript

We added the following sentences to the discussion.

“The notable sensitivity of G12A cells to dual GS/GLS inhibition, despite their low expression of nuclear FOXO1 and GS, likely stems from their low basal GS activity coupled with a KRAS G12A-driven reliance on GLS for processing exogenous glutamine to meet heightened metabolic demands. This highlights that different KRAS alleles can create vulnerabilities within the glutamine-glutamate cycle through varied mechanisms, broadening the potential of this therapeutic approach beyond just G12D/V mutants.”

Point-by-point response

Referee #1:

Ber et al. characterized a KRAS mutant (G12) isogenic SW48 cell line panel to study possible differences in functional/phenotypic outcome of these specific mutations. They conclude that G12D and G12V mutations most significantly link to changing glutamine/nitrogen metabolism through regulation of FOXO1. From a general (clinical) perspective it is of interest to see whether all KRAS mutations are in the end equal in terms of mediating tumorigenesis or that specific mutations show distinct features to an extent this may eventually guide mutation specific treatment. The availability of small molecules targeting G12C specifically lends credit to the idea of mutation specific treatment, albeit that this is more due to the biochemical nature rather than the specific biology that come along with this mutation.

Major issues.

As far as I can see for each mutation one cell line is being analyzed. If this is correct one wonders what the variability in outcome is when testing 3 biological replicates so three independently established cell lines with G12V; G12D etc. This multiplies the amount of work I realize this, but on the other hand I have no clue as to how much of the sometimes small difference between cell lines is due to clonal effects or not. Same holds for the metabolomic analysis as in the figure legends it is indicated that these concern 5 technical replicates. This is nice but one really wants to see biological replicates.

In addition to the following comment, please see also **Note #1** about the use of a single clone of isogenic cell lines.

Clarification of biological vs technical replicates: Of course, we agree with the referee on the importance of biological replicates. For metabolomics experiments (e.g., Fig. 2A,B, Fig. 3H, Fig. 4A,B,E), the "5 replicates" mentioned (e.g., for ¹³C-glucose tracing) refer to cells plated into five separate wells/dishes for each cell line, which were then independently cultured, subjected to the experimental conditions (e.g. labelling, isotope labeling, inhibitor treatment), and subsequently harvested and extracted for LC-MS analysis. Given the large number of samples and the need to time treatments carefully, we believe this protocol is appropriate. We also note that several overlapping control experiments (Figures 2A, 3H, 4A-B, 4E S3C and File S2) return identical results. For example, enhanced glutamine synthesis from glucose in G12D and G12V (Figure 2A and 3H (DMSO control, and control DMSO in iFOXO1 experiment in S3C, File S2); as well as enhanced ammonia incorporation into glutamine synthesis in G12D and G12V and transamination reaction in G12D (Figure 4A and DMSO control of Figure 4E, File S2).

With respect to FOXO1 as mediating the effect on nitrogen metabolism it should be considered that FOXO expression and neither nuclear localization is decisive predictor of activity. FOXOs are primarily regulated through post translational modifications so at present it is only suggested that FOXOs are activated by G12V and G12D but mechanistically it remains unclear, also because the presented data on AKT the major regulator of FOXO activity is not very different between the cell lines in the sense that this would explain the specific control of FOXO activity by G12V and G12D and not the other mutants.

See **Note #2**.

I appreciate (and agree with) the authors argument that cells should be tested in culture media that better reflect the actual metabolic environment. However, be this the case as far as I know glutamine levels/concentration between interstitial fluid and plasma do not differ that much so glutamine deprivation appears unlikely under physiological conditions (see e.g. Doglioni et al. PMID:39743589). Interstitial measurements indeed show substantial glucose deprivation but the condition tested here (2mM of glucose) is still well above the Km of the GLUT1 transporter so although reduced compared to the 25mM that many use it is still more than sufficient and not limiting.

See **Note #3**.

To me the rationale (if any) why (to solve what metabolic limitation) the G12D and V cell lines start to make glutamine out of glucose is not explained or clear. The use of GS/GLS inhibitors is also not make things easier to understand, because only a combo is really effective, but completely inhibiting glutamine metabolism is probably toxic to almost every cancer cell line irrespective of their mutational status.

Rationale for glutamine synthesis from glucose in G12D/V cells.

The enhanced synthesis of glutamine from glucose in KRAS G12D/V cells under nutrient stress is likely a multifaceted adaptation. We propose it serves to: i) **Manage nitrogen and detoxify ammonia.** These cells show higher intracellular ammonia (Fig. S5D). The FOXO1-driven upregulation of glutamine synthetase (GS/GLUL) (Fig. 3G&H) facilitates ammonia assimilation into glutamine, a crucial nitrogen source. ii) **Support biosynthesis.** This *de novo* glutamine supports sustained nucleotide synthesis (evident in Fig. S2B,D) and other biosynthetic needs. iii) **Confer a fitness advantage.** This metabolic rewiring provides a growth advantage under low glucose/serum conditions (Fig. 1E) and in low glucose/high glutamine environments (Fig. 2C,D), which can mimic aspects of the tumour microenvironment.

Interpretation of GS/GLS Inhibitor effects and selectivity.

We agree that high-dose inhibition of glutamine metabolism can be broadly toxic. Our key findings, however, highlight i) **Synergistic and selective killing.** We demonstrate a strong synergistic effect between MSO (GS inhibitor) and CB839 (GLS inhibitor) at lower doses (Fig. 5A,B right panels; Fig. S5B). This combination preferentially kills KRAS mutant cells over wild-type SW48 cells, particularly under low nutrient conditions where their metabolic reprogramming is pronounced (Fig. 5D). ii) **KRAS G12D-allele dependence.** The G12D-specific inhibitor MRTX1133 abrogates this enhanced sensitivity in SW48 G12D cells (Fig. S5E), linking the vulnerability directly to the oncogenic allele. iii) **Contrast with general toxicity.** This selective approach contrasts with the general toxicity observed with direct FOXO1 inhibition (AS1842856), which affected all cell lines (Fig. S5A) including the parental ones.

We propose that G12D/V cells, due to their FOXO1-driven metabolic reprogramming, become co-dependent on both glutamine synthesis and utilisation. Dual inhibition creates a synthetic lethal-like state more pronounced in these adapted mutant cells.

Changes to the manuscript

In the discussion, we added a paragraph in response to Referee 2, point 5 following the discussion about ammonia which should also address this question.

*“The enhanced fitness of G12D and G12V cells in low glucose and high glutamine (**Figure 2C, D**), fueled by their increased capacity for *de novo* glutamine synthesis from glucose, highlights their specific metabolic reprogramming. Conversely, the decreased viability observed in wild-type, G12A, and G12C cells under these same conditions when glutamine concentrations exceeded 1 mM (**Figure 2C**) may reflect their limited capacity to manage the byproducts of high glutamine flux, such as ammonia, when other nutrients are scarce. Unlike G12D/V cells, which upregulate the FOXO1-GLUL axis and demonstrate enhanced ammonia recycling (**Figures 3 and 4**), these cell lines might be more susceptible to ammonia toxicity or other metabolic imbalances induced by excessive glutamine catabolism in a nutrient-stressed state.”*

Hidden in supplementary S6 is validation of data on a different cell line (LIM1215) however here FOXO1 expression is increased in G12C and not G12V so in contrast to the SW48 panel and also nuclear staining is visible in all cell lines G12A and G12C as well. So to me argues that at least part of the observations are prone to biological variation between cell lines harboring the same mutation and iterates the suggestion to test more biological replicates.

LIM1215 data and biological variation: We will integrate LIM1215 data (Fig. S6) more prominently. The revised Discussion will address this inter-line variability while emphasising the conserved sensitivity of KRAS mutant cells in both backgrounds to dual MSO/CB839 inhibition, highlighting a common vulnerability despite upstream variations.

A more detailed description of the changes is included in note #1, together with a more general response about biological replicates.

Taken together interesting study but to me it is unclear how a specific KRAS mutation links to regulation of FOXO1 and how this then translates to a mutation specific (metabolic) phenotype. The data suggest to me that the link is quantitative rather than qualitative, so e.g. a two fold increase in G12C would make this mutant comparable to G12V. The part on regulation of metabolism is at present too descriptive and also does not provide mechanistic insight as to how these observations link to the specific KRAS mutations and not to the other ones, so the conclusion stated in the abstract "KRAS mutations at codon 12 induce allele specific metabolic programs" is to me not supported by the data in the sense that the authors do not show that the other alleles are not inducing these metabolic programs if expressed at e.g. a slightly higher level.

A simple dosage/KRAS:GTP level model doesn't fully explain our distinct metabolic phenotypes. For instance, SW48 KRAS G12V shares metabolic commonalities with G12D, while G12A does not consistently exhibit the same FOXO1 phenotype, despite varying active KRAS levels (Fig S1A, Figs 1-4). This aligns with known biochemical differences among KRAS mutants (e.g., Hunter et al., 2015). However, we agree the abstract's phrasing "allele specific metabolic programs" could be considered an overstatement and has to be refined.

Proposed changes to the manuscript.

We have revised the text (see tracked document) to avoid a similar phrasing, in addition to the following changes.

Graphical abstract.

Previous text: "KRAS mutations at codon 12 induce allele-specific metabolic programs reprogramming in colorectal cancer cells. Ber et al. reveal that G12D and G12V mutations promote FOXO1-dependent glutamine and nitrogen recycling, uncovering a therapeutic vulnerability to dual inhibition of glutamine synthetase and glutaminase."

Amended text: "KRAS mutations at codon 12 induce profound metabolic changes in colorectal cancer cells. Ber et al. reveal that G12D and G12V mutations promote FOXO1-dependent glutamine and nitrogen recycling, uncovering a therapeutic vulnerability to dual inhibition of glutamine synthetase and glutaminase."

Discussion section.

Previous text: "Although the specific mechanisms by which similar oncogenic KRAS alleles cause different phenotypes are not fully established, different mutations at glycine-12 and -

13 exhibit altered GTP hydrolysis, differential engagement of effector proteins and signalling and signalling dynamics. ”

Amended text: *"Also the overall levels of active KRAS are important, but our data suggest that a simple dosage model does not fully explain the distinct metabolic phenotypes observed across the G12 mutants. For example, in SW48 cells, KRAS G12V shares key metabolic reprogramming features with G12D, despite potentially different levels of GTP-bound KRAS, whereas G12A, even with comparable active KRAS levels to some other mutants (Figure S1A), diverges in its downstream FOXO1 engagement. This aligns with biochemical studies demonstrating intrinsic differences in GTP hydrolysis, effector engagement, and signalling dynamics among KRAS G12 variants. These inherent allele-specific biochemical properties likely contribute to the qualitative and quantitative differences in the resulting metabolic landscapes and underpin the observed variations in FOXO1 expression, nuclear localisation, and overall functional state.*

Referee #2:

In the present manuscript, the authors reveal how specific KRAS G12 mutations influence cellular metabolism in colorectal cancer. Using isogenic cell line models combined with transcriptomic, metabolomics and biochemical approaches, they demonstrate that different KRAS alleles induce distinct metabolic programs affecting glutamine metabolism and nitrogen recycling. The study highlights that FOXO1 is upregulated in KRAS G12D and G12V mutant SW48 cells, enhancing cell viability under low-nutrient conditions. Furthermore, the authors showed that simultaneous inhibition of GS and GLS can significantly reduce cell viability of certain KRAS mutant cells.

The manuscript has some interesting and potentially important findings, but there are a number of key issues that need to be addressed.

Major:

1) While the use of an isogenic colorectal cancer cell line harboring heterozygous mutations in KRAS provides a valuable and controlled system to study the functional effects of specific KRAS mutations, the exclusive reliance on these models limits the translational value of the conclusions. Such panels may not fully reflect the biology of the targeted cancer pathways. Even within the manuscript, the authors show that KRAS mutations in the SW48 panel do not affect downstream pathways, and experiments using the LIM1215 line did not fully reproduce the results. It should be noted that neither the SW48 nor the LIM1215 cells are "normally" mutant for KRAS. These mutations were introduced artificially, and it isn't clear if the cells even rely on KRAS to maintain their signaling structure.

To strengthen the impact and biological relevance of the study, it is recommended to repeat key experiments in one or more additional cell lines that harbor endogenous KRAS mutations.

See **note #1**.

2) There is a lack of statistical analysis for western blots throughout the manuscript. It is recommended to quantify band intensity and normalize to a loading control.

We agree on the value of Western Blot quantification. Given that we have already several panels per figure and also several supplementary figures, we provide the quantification as additional supplementary file. We confirm that all our observations are confirmed and, therefore, we had only added a sentence in Material and Methods:

*“Western Blots were quantified with Image Studio, and the quantification is provided in **File S3.**”*

The only exception is Fig. S6D. The enrichment of FOXO1 is not statistically significant, albeit the observed trends are present in all repeats. This is likely caused by the quality of the fractionation, which we could not improve. However, we have now included the quantification of the immunofluorescence data as well, showing similar trends but with sufficient statistical power (see addition in Fig. S6C).

3) The rationale behind the choice of full media or low-nutrient media conditions is unclear in some cases. It would be helpful to explain why each condition was chosen. For instance, the authors demonstrate modest upregulation of the ERK pathway in low-nutrient conditions for G12D-mutant cells; however, they study the effects of MRTX1133 under full media conditions.

We appreciate the referee's point regarding media conditions. Briefly, the isogenic panel of SW48 cell have been extensively used in the past by other groups and characterised in full media. Therefore, the experiments we performed in full media were key to comparing our work with that of others and for the initial investigation that led us to discover a link between different mutants and FOXO signalling. Once these two points were established, we determined optimal conditions to study the metabolic phenotype, *i.e.*, conditions that are not designed to stimulate the proliferation of cell lines (high glucose, high serum) but more representative of tumour interstitial fluid, as explained in **note #3**.

On the use of MRTX1133. Our manuscript presents experiments with MRTX1133 in both full media and low-nutrient conditions to assess its impact on the KRAS G12D-FOXO1-GLUL axis.

- Figure 3C and associated text describe the effects of MRTX1133 on FOXO1 and AKAP12 protein expression in full media.
- Figure 3D and associated text show MRTX1133 effects on FOXO1, AKAP12, and crucially GS (GLUL) protein expression in low-nutrient conditions.
- Figure S3B further details mRNA expression of FOXO1 and GLUL with MRTX1133 treatment in both full and low-nutrient media.
- The initial characterization of MRTX1133's effects on direct downstream signaling (pERK, pAKT; Fig S1D, S1E) was performed in full media. This was to first validate the inhibitor's specificity and its ability to engage the KRAS G12D target protein under standard culture conditions, which is a common approach before exploring effects under more specialized or stress conditions.

The modest ERK pathway upregulation in low-nutrient conditions for G12D cells (Fig S1C) prompted our deeper investigation of MRTX1133's impact on the FOXO1-GLUL axis specifically in these more stringent conditions (Fig. 3D, Fig. S3B), where the metabolic phenotypes are most pronounced. In the revised manuscript, we ensured that the rationale for transitioning between these conditions for different experimental questions is clearly articulated and that experimental conditions for each panel presented are explicitly defined both in the main text and captions (see **note #2**).

4) In the first paragraph of the results in chapter three, the authors state: "Intracellular metabolomics showed that TCA (tricarboxylic acid) cycle metabolites like α -ketoglutarate and succinate are

significantly lower in G12D and G12V mutant cells (Figure 2A)." However, Figure 2A does not show a significant difference in α -ketoglutarate levels between WT and G12D cells.

We thank the referee for their careful attention to Figure 2A. Upon review, the referee is correct that for the displayed α -ketoglutarate m+2 isotopologue, the levels in G12D cells are not significantly lower than in WT cells. This discrepancy was caused by an incorrect revision of the text and figure meant to make the manuscript more concise.

The text in the discussion referring to Fig. 2A was still correct "Metabolic flux analysis performed with ^{13}C -glucose shows that all mutant cell lines avidly consume pyruvate to feed the TCA cycle, as shown by high levels of labelled citrate, which is eventually utilised to maintain high glutamate to support biogenesis in mutant cells (Figure 2A)."

We amended the text and Fig. 2A accordingly.

Former text: "Intracellular metabolomics showed that TCA (tricarboxylic acid) cycle metabolites like α -ketoglutarate and succinate are significantly lower in G12D and G12V mutant cells (Figure 2A), suggesting a lower TCA cycle activity. Surprisingly, G12D, G12V, and G12C to a lesser extent, exhibited a substantial flow of ^{13}C -glucose carbon towards glutamine via glutamate, as indicated by the presence of m+2 isotopologues."

Amended text and figure: "Intracellular metabolomics showed that the fluxes of ^{13}C -glucose carbon in the TCA (tricarboxylic acid) cycle are different across mutant cell lines (Figure 2A). For example, citrate m+2 is higher than wild-type in all mutant cell lines, fuelling their *de novo* glutamate production. However, while the G12D, G12V and G12C direct ^{13}C -glucose carbon flux towards glutamine synthesis, the G12A mutant maintains a higher flux through the TCA cycle and high glutamate levels"

5) Can the authors explain the significant decrease in viability of G12A/C and especially WT cells when glutamine concentration is increased (Figure 2C)?

This is an interesting observation that highlights the differential metabolic adaptations among the KRAS genotypes under nutrient stress. We propose that the decreased viability of wild-type (WT), G12A, and G12C cells at high glutamine concentrations (above 1 mM) specifically under low glucose (2 mM) and low serum (1%) conditions (Figure 2C, bottom panel) could be due to an inability to efficiently manage the metabolic byproducts of high glutamine flux when other key nutrients are scarce.

High levels of glutamine are catabolised by glutaminase (GLS) to glutamate and ammonia. While KRAS G12D and G12V cells demonstrate enhanced FOXO1-dependent glutamine synthetase (GS/GLUL) expression and increased ammonia recycling capabilities (**Figures 3-4**), the WT, G12A, and G12C lines may lack this capacity. Consequently, in these cells, the ammonia generated from excessive glutamine catabolism under reduced nutrients might accumulate to toxic levels, leading to reduced viability. Our data show G12D/V cells are proficient in incorporating ammonia (**Figure 4A**). Moreover, under low glucose conditions, cells become more reliant on alternative substrates like glutamine for energy and biosynthesis. An excessive influx of glutamine in cells not fully adapted to efficiently channel its products (like WT, G12A, and G12C) could lead to other metabolic imbalances, contributing to reduced cell fitness.

We have added a short description in the amended discussion.

“The enhanced fitness of G12D and G12V cells in low glucose and high glutamine (Figure 2C, D), fueled by their increased capacity for de novo glutamine synthesis from glucose (Figure 2A, B, 3H), highlights their specific metabolic reprogramming. Conversely, the decreased viability observed in wild-type, G12A, and G12C cells under these same conditions when glutamine concentrations exceeded 1 mM (Figure 2C) may reflect their limited capacity to manage the byproducts of high glutamine flux, such as ammonia, when other nutrients are scarce. Unlike G12D/V cells, which upregulate the FOXO1-GLUL axis and demonstrate enhanced ammonia recycling (Figures 3 and 4), these cell lines might be more susceptible to ammonia toxicity or other metabolic imbalances induced by excessive glutamine catabolism in a nutrient-stressed state.”

6) *The authors used MRTX1133 to test the dependency of FOXO1 expression on oncogenic KRAS, focusing only on G12D cells. It would be worthwhile to study how FOXO1 expression changes in response to KRAS inhibition or knockout in other KRAS-mutant cells.*

See **Note #1**.

7) *"FOXO1 inhibition reduced the viability of all tested cell lines, particularly cells with lower nuclear FOXO1, including KRAS wild-type cells." Did the authors study how inhibition of oncogenic KRAS and FOXO1 together affects the viability of KRAS-mutant cells?*

This is an interesting point that we had not explored. Our strategy, upon establishing the KRAS G12D-FOXO1 link (Figures 3C, 3D), was to investigate whether targeting the downstream metabolic pathways regulated by FOXO1 (*i.e.*, glutamine synthesis and catabolism via GS/GLS) could offer a more selective therapeutic vulnerability for KRAS mutant cells. This led to the findings with MSO and CB839 (Figure 5). MRTX1133 became available only in the late stages of our work and was thus used to test our hypothesis that the mutant alleles are responsible for FOXO1 upregulation. Given that our work is already quite broad and complex, we preferred to focus on downstream effects.

8) *"While we observed a higher sensitivity of KRAS mutant cells relative to wild-type SW48 cells, the effects were only significant at high MSO concentrations (Figure 5A, left panel)". However, it is difficult to distinguish any difference in sensitivity from the curves, the error bar in the WT group appears quite high.*

We agree with the referee. Although there is statistical significance at the highest concentrations, we removed that statement as it was non-essential and might be distracting from the main observations.

9) *The G12A cells show much higher sensitivity to the combination of GLUL and GLS inhibitors than G12D/V cells, especially under low nutrient conditions. This suggests that targeting these two enzymes may be more promising in treating tumors with KRAS G12A mutations than focusing on G12D/V-mutant cells.*

See **note #4**.

10) *"Knocking down the two genes by siRNA also confirmed the sensitivity of G12D and G12V mutant cells to the inhibition of GLS and GS together (Figure S5G)." The authors probably meant Figure S5C. Since G12A cells respond better to the inhibition of GLUL and GLS, it would be worthwhile to study how knockdown of these genes affects the viability of G12A cells.*

We thank the referee for pointing out these details.

Figure reference typo. The referee is correct. The reference to "Figure S5G" on page 10 is a typographical error. It should indeed refer to Figure S5C, which shows the siRNA knockdown data for GLUL and GLS in WT, G12D, and G12V cells.

The focus on G12A mutation has been discussed in **note #4**.

11) *"To mitigate the risk that the observed sensitivities entirely depend on the SW48 cell line background, we repeated crucial experiments using a second isogenic panel derived from the colorectal cancer cell line LIM1215 (Figure S6). The LIM1215 panel harboured the same mutant alleles, also exhibited upregulation of FOXO1, and revealed the same sensitivity of mutant LIM1215 to combinatorial drugging of glutamine synthetase and glutaminase, as observed in SW48 isogenic panel."*

It is recommended to expand this part of the manuscript. Due to the poor quality of the western blots, it is difficult to draw conclusions about FOXO1 levels between WT and mutant cells. Also, Bliss analysis patterns differ between the SW48 and LIM1215 lines.

See **note #1**.

Regarding BLISS analysis, although the absolute values change, the sensitivity of the combination treatment in mutant cells over WT cells is consistent.

Minor:

I suggest the authors consider expanding the introduction by focusing on metabolic pathways that were studied in the manuscript.

While appreciating the comment, we have opted to do only minor changes to the introduction to avoid anticipating too much from the results and discussion. We added a sentence:

"The overarching role of KRAS in these metabolic shifts is appreciated¹, but a detailed understanding of how distinct mutations modulate these specific pathways (e.g., glutamine utilisation, de novo synthesis, and nitrogen/ammonia handling) and the regulatory networks involved remains an area of active investigation."

To better frame out work and slightly expand the introduction to the key pathways we then discuss in depth later in the manuscript. We have also refined the concluding remarks:

“We show that the FOXO1-GLUL axis upregulates glutamine metabolism in these cells, leading to enhanced glutamine synthesis from extracellular glucose. At the same time, the upregulation of FOXO signalling, a pathway so far primarily associated with apoptosis in cancer^{2–6}, enhances ammonia recycling via glutamine synthesis and transamination pathways, supporting the survival advantage of G12D and G12V mutant cells. Notably, we identified that the simultaneous targeting of glutamine synthesis and glutaminolysis selectively kills G12-mutant KRAS cell lines in two heterozygous isogenic colorectal cell lines (SW48 and LIM1215) compared to their wild-type counterpart, suggesting a high dependency of KRAS mutant cells on nitrogen recycling and a possible new venue for therapeutic intervention.”

To focus the readers' attention on the two main findings, *i.e.* the unique FOXO1 signature in G12D and G12V mutant cells, and the common sensitivity to inhibition of the glutamate-glutamine cycle shared between all mutants and both isogenic panels.

Referee #3:

The study by Ber et al reports the results of a comprehensive transcriptomic and metabolomic characterisation of an isogenic cell line panel derived from SW48 colorectal cancer (CRC) cells harbouring different mutants of the Ras oncogene (V12G, V12D, V12C and V12A compared to WT). The authors initially perform RNA sequencing across the cell line panel to identify deregulated gene signatures linked to Ras mutations with high prevalence in CRC (V/D compared to G/C). This revealed that FOXO1 signalling is induced in D/V mutants. The authors also perform metabolomic characterisation using ¹³C-glucose tracing to show altered glutamine metabolism. They expand the analysis of FOXO1 regulation by showing enhanced nuclear localisation in D/V cells. This was most prominent in G12D and G12V mutants. The authors next implicate FOXO1 in the regulation of glutamine metabolism using a FOXO1 inhibitor. The authors then move on to using ¹⁵N-glutamine tracing to reveal that Ras regulates ammonia recycling via FOXO1-dependent regulation of GS. Finally, they show that combined inhibition of GS and GLS limits survival of Ras mutant cancer cells and this is enhanced by low nutrient conditions. The authors conclude that targeting ammonia recycling could be a therapeutic target in Ras mutant cancers.

While the manuscript contains some interesting findings, its overall quality is quite difficult to assess. The data are presented in a highly confusing manner, with the figures not being discussed in the correct order and jumping from main to supplementary data without clear reason. While most of the experiments were done in all cell lines, the investigation of altered ammonia metabolism focusses only on the G12D and G12V mutants, making it difficult to connect to the inhibitor results. The manuscript contains extensive supplementary material, some of it being unrelated to the main line of investigation. The discussion either reiterates the results section or overstates the findings and requires substantial editing for clarity. In addition, page and line numbers are missing making it hard to conduct the review. Despite these major limitations, the manuscript reports some interesting findings, most importantly the somewhat unexpected observation that mutant Ras activates FOXO1. However, the authors need to address the inconsistencies in the data to provide sufficient support for their conclusion before the work can be published.

Major points:

1) The initial cell line characterisation is confusing. Why do the Ras mutant cells not show activation of ERK and AKT signalling? The experiment shown in S1B and C should have been performed after growth factor deprivation. S1D uses 1% FCS but also reduced glucose (2 mM), which could affect the activity of these signalling pathways. As all other experiments built on these Ras mutations, it is mandatory that their functionality is proven.

We respectfully disagree with the characterization that our initial cell line analysis is confusing or that the functionality of the KRAS mutations is not proven. We would like to clarify as follows:

- **Activation of KRAS protein:** Figure S1B (now S1A) clearly demonstrates through GTP-RAS immunoprecipitation that all engineered mutant KRAS cell lines (G12A, G12C, G12D, G12V) exhibit upregulated levels of active, GTP-bound KRAS compared to the parental wild-type SW48 cells when cultured in full media. This directly confirms the biochemical functionality of the introduced mutations in terms of enhancing KRAS activity. The two isogenic panels we used, SW48 in particular, have been repeatedly characterised and used by numerous groups and our initial characterisation presented in Fig. S1 is consistent with that of others.
- **Downstream ERK/AKT signaling (Full Media - Fig S1C):** Our manuscript acknowledges that this increase in active KRAS did not translate into a clear, uniform upregulation of pERK or pAKT across all mutant lines in full media conditions (Figure S1C, now S1B). This is an important biological observation. As noted by Hood et al. (2019) and others, RAS signaling can be isoform-specific and growth factor dependent, and downstream pathway activation is often subject to complex regulatory feedback that can temper overtly strong signals. We have amended the following sentence: “Lack of hyper-activation of these pathways is congruent with prior observation both in cell lines and tumours, particularly in the context of heterozygote mutations” and new references (doi: 10.1158/1078-0432.CCR-17-3438 and 10.1158/0008-5472.CAN-07-0108).
- **ERK/AKT signaling (Low Nutrient/Serum - Fig S1D, now S1C):** We then assessed signaling under conditions of reduced serum (1% FCS) and glucose (2mM). In this combined stress condition (Figure S1D, now S1C), a modest upregulation of the ERK pathway became more apparent for the G12D and G12V cell lines. This suggests these specific mutants might be more capable of sustaining this signaling cascade under limited nutrient and growth factor availability, which is a relevant phenotype.
- **Functional link of KRAS G12D to phenotypes:** Crucially, for the KRAS G12D allele, its direct functional role in driving downstream signaling and the key phenotypes discussed in our manuscript is robustly demonstrated using the G12D-specific inhibitor MRTX1133. Treatment with MRTX1133:
 - Decreases pERK and pAKT in SW48 G12D cells (Figure S1E, now S1D, performed in full media to first validate target engagement).
 - Reduces FOXO1 expression and its target GS/GLUL (Figure 3C in full media; Figure 3D in low nutrients).
 - Abrogates the sensitivity to the MSO/CB839 drug combination (Figure S5E). These results unequivocally prove the functionality of the KRAS G12D mutation in mediating the specific effects central to our study.

In summary, our initial validation of the cell line are consistent with results already presented in the literature, with the outcome expected by an isogenic line not overexpressing mutant KRAS but expressing a single copy of a mutant allele.

2) A major limitation of the study is that the experiments switch between full media and low nutrient conditions. The exact formulation of the low nutrient medium (2 mM glucose, 1% FCS) is quite arbitrary, despite the claims made in the discussion that this may represent the in vivo condition, and combines glucose withdrawal with reduced availability of serum-derived growth factors and lipids.

Nevertheless, the "low nutrient" condition reveals the more interesting phenotypes, suggesting that all experiments should have been done like this.

Our initial experiments were performed in full media for two main reasons: first, to validate the cell lines and allow comparison of our results with existing literature (see also response to point #1); second, our initial hypothesis was that individual alleles might cause differences, the analysis of which could help us identify either common or specific vulnerabilities. Once we identified a key metabolic phenotype, we then designed experiments specifically tailored to investigate it under appropriate conditions.

The rationale for the specific formulation of our low-nutrient medium and its physiological relevance will be further clarified in the revised manuscript text (as per 'Note #3' in our revision plan, referencing manuscript changes detailed in our response to Referee #1, Concern 3).

We also revised all figures to ensure that the specific media used is clearly indicated.

3) The mechanism by which mutant Ras induces FOXO1 activation remains unclear. While the increased nuclear localisation seems highly relevant (Fig. 3E and F), the authors also show some minor regulation of FOXO1 mRNA (Fig. S3B). A more detailed investigation of this mechanism would greatly enhance the impact of the study.

See **note #2**.

4) The authors claim that oncogenic Ras regulates expression of GLUL/GS via FOXO1. However, inhibition of Ras using MRTX1133 does not reduce GS expression in full media (Fig. 3C), while the induction is only seen in G12D cells in the low nutrient condition. Did the inhibitor work at all? Similarly, siRNA-mediated deletion of FOXO1 only had a minor effect on GLUL mRNA expression, and baseline GLUL expression did not follow the suggested pattern of induction by the two prevalent Ras mutations (Fig. S3E).

MRTX1133 and GS/GLUL Expression (Fig 3C vs. 3D):

- The referee is correct that MRTX1133 (100 nM) did not reduce GS/GLUL protein expression in SW48 G12D cells in **full media** (Fig. 3C). However, MRTX1133 treatment did significantly reduce GS/GLUL protein expression in SW48 G12D cells under **low-nutrient conditions** (Fig. 3D).
- This context-dependent effect is an important finding. It suggests that the reliance of KRAS G12D on FOXO1 to maintain GS/GLUL expression is more critical under metabolic stress.
- The inhibitor's efficacy (target engagement) is demonstrated by the consistent reduction of AKAP12 and FOXO1 protein by MRTX1133 in G12D cells under both full media (Fig. 3C) and low-nutrient conditions (Fig. 3D), as well as its effects on pERK/pAKT (Fig. S1E-F, now S1D-E).

siRNA-mediated knockdown of FOXO1 and GLUL mRNA (Fig S3E):

- The baseline GLUL mRNA expression in Figure S3E (siControl bars) does show higher levels in G12D compared to WT cells under low-nutrient conditions, which is consistent with the pattern of induction by these prevalent KRAS mutations seen at the protein level (Fig 2F, Fig 3D DMSO lanes) and RNAseq (Fig 2E).
- Figure S3E shows that siRNA-mediated knockdown of FOXO1 in SW48 G12D cells cultured in **low-nutrient conditions** did lead to a reduction in GLUL mRNA . While the

effect might appear modest for G12D in that particular representation, it was consistently observed.

4) [5] *The observation that combined inhibition of GS and GLS reduces viability of mutant Ras cells is interesting. However, this effect seems to be strongest in G12A cells, which only show minor activation of FOXO1 and GS. How does this square up with the conclusions drawn by the authors that this could be a strategy to eliminate mutant Ras cancers?*

See **note #4**.

Minor

points:

- *The Waddington scheme used in the graphical abstract implies that the study investigated "metabolic trajectories" induced by different Ras mutants, which is an overstatement. This should be removed.*

Graphical abstract amended

- *Fig. S1A is difficult to understand and not relevant for the study.*

Fig. S1A was removed.

Page 7 middle paragraph should be "An increase in nuclear FOXO1...".

All cells enhance expression in the cytoplasm, but only some line increase nuclear localisation. The original text was correct.

- *The statical analysis in Figure 5D should be checked. Only biologically independent replicates (n=4) should be used for the calculation.*

We agree and have changed this panel and a similar one in Figure S5.

- *The limitation section raises some valid points but also includes several statements that merely describe standard procedures to assure robustness of the data. Using sufficient replicates is just good scientific practice but does not support the conclusions.*

We agree and have completely amended the section (see **Note #1**)

- *Showing individual tissue cores copied from the Human Proteome Atlas is very arbitrary and should not be included (Fig. S7D).*

We removed panel S7D.

Additional changes

Given the large number of panels, we include in the submission, we have done a full check for consistency between the data and the panels presented. Fig. 3H and 4E (asparagine) were replotted because were incorrectly generated from different datasets. The interpretation of the data does not change. The data sources we had originally provided are unaltered as the error was done only during the assembling of the panels. We have also

KRAS drives FOXO1–GLUL axis

edited the data availability statement while the data is getting processed by Gene Expression Omnibus and Metabolomics Workbench.

Dear Dr. Esposito,

Thank you for the submission of your revised manuscript to our editorial offices. I have now received the reports from the three referees that I asked to re-evaluate the study, you will find below. As you will see, referees #1 and #3 now support publication of your study in EMBO reports. Referee #2, however, indicates that a major issue has not been addressed and requests further revision. After cross-commenting with the other two referees, who nevertheless support publication, I have decided to proceed with the manuscript. But it will be necessary to address the central remaining issue of referee #2 in the final manuscript text, clearly mentioning and discussing this limitation. Moreover, referee #3 has a minor comment I also ask you to address in a final revised manuscript, as well as the editorial requests below. Please also provide a final p-b-p-response to the remaining referee points and the editorial requests.

Editorial requests:

- Please add up to five keywords to the manuscript and order the sections like this, using only these names: Title page - Abstract - Keywords - Introduction - Results - Discussion - Methods - Data availability section - Acknowledgements - Disclosure and Competing Interests Statement - References - Figure legends
- We now use CRediT to specify the contributions of each author in the journal submission system. CRediT replaces the author contribution section. Please use the free text box to provide more detailed descriptions and do NOT provide your final manuscript text file with an author contributions section. See also our guide to authors: <https://www.embopress.org/page/journal/14693178/authorguide#authorshipguidelines>
- The data availability section (DAS) is restricted to information about large datasets generated in the study that have been deposited externally. Please provide here only access information for the RNA-seq. and metabolomics datasets and direct links. Please remove now all referee tokens and make sure that the datasets are public latest upon online publication of the manuscript. Please remove the sections 'Lead contact' and 'Materials availability' from the DAS.
- Please name the 'Supporting information' file 'Appendix' and provide it as pdf file. The Appendix should have page numbers and needs to include a table of content (TOC) on the first page (with page numbers). Please state on the first page 'Appendix for ...' followed by title, with the TOC below. Please follow the nomenclature 'Appendix Figure Sx' and 'Appendix Table Sx' throughout the text (for the callouts) and also label the figures and tables in the Appendix file according to this nomenclature.
- There are two Supplemental ZIP folders (S1 and S2) uploaded. It seems these contain source data, i.e. data and information used to create figure panels. Thus, please add this information to the source data files for the respective figures. In case, please upload these as source data for Appendix figures (one folder - see below).
- Regarding Supplemental File S3, please add these quantifications to the Western blot panels in the indicated figures. There are no space limitations, and I do not see a reason why the quantifications should be provided as Supplement.
- Please confirm that for all Western blot panels (main and Appendix figures) the loading control was run on the same gel as the other proteins detected. Please note that we discourage comparisons between samples on different gels/blots, even if the samples derive from one experiment, as confounding factors reduce comparability. If unavoidable, the figure legend must state that the samples derive from the same experiment and that gels/blots were processed in parallel. If a 'representative' loading control is shown for multiple gels/blots, the intra-gel controls should be shown in the source data files, and the figure legends should describe the data displayed accurately. See our author guidelines:

<https://www.embopress.org/page/journal/14693178/authorguide#datapresentationformat> (section 'Electrophoretic gels and blots').

and

<https://www.embopress.org/image-integrity>

Thus, please also provide the source data files (uncropped images) for the Western blots shown in the Appendix figures. Please upload these as one ZIPed folder containing all the source data for the Appendix figures in separate folders.

- Most importantly: Please add molecular weight markers to all Western blots shown!
- Please check again that the number "n" for how many independent experiments were performed, their nature (biological versus technical replicates), the bars and error bars (e.g. SEM, SD) and the test used to calculate p-values is indicated in the respective figure legends (main and Appendix figures). Please also check that all the p-values are explained in the legend, and that these fit to those shown in the figure. Please provide statistical testing where applicable. Please avoid the phrase

'independent experiment' but clearly state if these were biological or technical replicates. Please also indicate (e.g. with n.s.) if testing was performed, but the differences are not significant. In case $n=2$, please show the data as separate datapoints without error bars and statistics. See also:

<http://www.embopress.org/page/journal/14693178/authorguide#statisticalanalysis>

If $n < 5$, please show single datapoints for diagrams. It seems presently 'n.s.' is not indicated in most diagrams. Moreover:

- Please note that the exact p values are not provided in the legends of figures 2A-E; 3B, F, G, H; 4A, B, E; 5D.
 - Please note that the box plots need to be defined in terms of minima, maxima, centre, bounds of box and whiskers, and percentile in the legends of figures 4A, B, E; 5D.
 - Please note that information related to n is missing in the legend of figure 3G.
 - Please note that the error bars are not defined in the legend of figure 1E.
 - Please note that the measure of center for the error bars needs to be defined in the legend of figure 1D.
- Please add to each legend (main and Appendix figures, where applicable) a 'Data Information' section explaining the statistics used or providing information regarding replicates and scales. See:

- Please remove the Reagents & Tools Table from the main manuscript file and upload it as separate file using the provided template:

<https://www.embopress.org/page/journal/14693178/authorguide#structuredmethods>

Please add callouts to the R&T table to the methods section where appropriate.

- Please use our reference format:

- Please provide a fully completed author checklist (with author names, manuscript number and journal name). Please update the callout to Fig. S7 in the last row. The re-used data needs to be cited in the reference list. See section 'Data citation' here:

<https://www.embopress.org/page/journal/14693178/authorguide#referencesformat>

- Please make sure that all the funding information is also entered into the online submission system and that it is complete and similar to the one in the acknowledgement section of the manuscript text file. Presently, the grant 10107542 (the Alexander von Humboldt Foundation in the framework of the Alexander von Humboldt Professorship endowed by the Federal Ministry of Education and Research) is missing in the submission system. Please check.

In addition, I would need from you uploaded separately (please remove this from the manuscript text file):

Please use this link to submit your revision: <https://embor.msubmit.net/cgi-bin/main.plex>

Best,

Referee #1:

Authors have carefully addressed the point raised in previous review round. I accept their line of reasoning and I think they have carefully rewritten their manuscript as to avoid over-statements. I understand that resolving the isoform specific regulation of FOXO was not deemed relevant by editors for publication in EMBO reports so this being the case I am OK (or better must be OK) with the answer of authors.

Referee #2:

The authors have improved the ms in many aspects, but my central issue remains. That is, while they verified their main points in two different isogenic cell lines, neither of these truly reflect KRAS biology. Neither the LIM1215 cells nor the SW48 cells are normally KRAS mutant. Rather, the mutations were artificially added to these cells. They are useful up to a point, but we need to see some confirmation outside of this artificial system. Analysis of a panel of CRC cells mutant for KRAS G12V vs G12D cells, for the main phenotype at issue, would strengthen the paper considerably.

Referee #3:

The authors have addressed the reviewers' comments mainly by altering the text and providing additional supplementary data. In particular, the authors added quantification of the western blot data, which overall support their conclusions. They also clarified some inconsistencies in the text. While some of the original issues still persist, the overall message of the manuscript is interesting and the reported data should provide a valuable resource for the community.

Minor comments:

The authors should re-evaluate all statements of "significance" in the text and figure legends. This term should only be applied if significance was tested.

It should be noted that the submission contains a very large amount of additional data and it is not feasible to fully evaluate these as part of the review process. It should be assessed whether all supplementary data files are needed.

Referee #1:

Authors have carefully addressed the point raised in previous review round. I accept their line of reasoning and I think they have carefully rewritten their manuscript as to avoid over-statements. I understand that resolving the isoform specific regulation of FOXO was not deemed relevant by editors for publication in EMBO reports so this being the case I am OK (or better must be OK) with the answer of authors.

We would like to thank you for your feedback and the opportunity to improve our work.

Referee #2:

The authors have improved the ms in many aspects, but my central issue remains. That is, while they verified their main points in two different isogenic cell lines, neither of these truly reflect KRAS biology. Neither the LIM1215 cells nor the SW48 cells are normally KRAS mutant. Rather, the mutations were artificially added to these cells. They are useful up to a point, but we need to see some confirmation outside of this artificial system. Analysis of a panel of CRC cells mutant for KRAS G12V vs G12D cells, for the main phenotype at issue, would strengthen the paper considerably.

We appreciate the overall positive comment and the observation on the limitations of our study, which is based on isogenic lines. However, highly heterogeneous CRC lines (non-isogenic panels) will provide additional limitations. To genuinely go beyond our current work, we will focus our future research on animal studies and patient-derived materials that are, however, outside the scope of this manuscript.

While we might partially disagree on the usefulness of additional experiments on other KRAS mutant CRC lines, we do agree on the need to interpret the results cautiously and within the constraints of specific experimental systems. We have thus fully revised our discussion of the limitation of our study; we have integrated it directly into the main text rather than providing it as a separate supplemental section. This clearly frame how our data should be interpreted:

“It is important, however, to acknowledge certain limitations in this study. Our conclusions are derived from engineered isogenic cell lines in which KRAS mutations were artificially introduced. We recognise that these in vitro models, based on single clones, cannot fully recapitulate the genetic heterogeneity and evolutionary context of tumours harbouring endogenous mutations. To mitigate this, we confirmed our central therapeutic vulnerability in a second, independent colorectal cancer background (LIM1215), demonstrating the phenotype is not a cell-line-specific artefact. Most critically, we established that the key phenotypes are directly dependent on the oncogenic allele itself by using the KRAS G12D-specific inhibitor MRTX1133 to reverse them. Nonetheless, these models cannot fully capture the complexity of three-dimensional tissues or systemic physiology, even when cultured in low-nutrient conditions designed to reflect aspects of the tumour microenvironment.

Despite these limitations, our work suggests that the glutamine-glutamate cycle provides a growth advantage to KRAS mutant cells with a functional role of the KRAS-FOXO1-glutamine metabolism axis. However, the precise upstream mechanisms linking each KRAS G12 variant to the differential regulation of FOXO1 were not exhaustively detailed. Therefore, the contribution of specific post-translational modifications, for example, remains an important area for future investigation. Taken together, our work suggests that the glutamine-glutamate cycle and KRAS-dependent modulation of FOXO signalling are potentially attractive new targets for validation studies in more clinically-relevant models and drug discovery.”

Referee #3:

The authors have addressed the reviewers' comments mainly by altering the text and providing additional supplementary data. In particular, the authors added quantification of the western blot data, which overall support their conclusions. They also clarified some inconsistencies in the text. While some of the original issues still persist, the overall message of the manuscript is interesting and the reported data should provide a valuable resource for the community.

Minor comments:

The authors should re-evaluate all statements of "significance" in the text and figure legends. This term should only be applied if significance was tested.

It should be noted that the submission contains a very large amount of additional data and it is not feasible to fully evaluate these as part of the review process. It should be assessed whether all supplementary data files are needed.

We would like to thank you for the opportunity to we had to improve our manuscript. We have revised all instances where we mention statistical significance and all figure legends to ensure we explicitly state which tests were performed and the p-value returned.

Full list of changes to figures.

To address editorial requests, referees' questions and to correct errors we have identified during the revision, we have amended several figures. The raw data and the interpretation of the data remains unaltered, but all changes are listed below for full transparency.

- We added quantifications of Western Blots and added molecular weights to panels 2F, 3A, 3C, 3D, S1B-E, S3A, S6D.
- After rerunning all statistical testing to report p values we noticed a few changes because of a different multiple comparison testing used on some experiment. We ensured now all statistical tests are consistent with material and methods. We amended the following panels: 3F,H, 4A, S2D, S3B,E, and S5E.
- Fig. 4E. We have noticed that one panel (asparagine) did not correspond to the actual data. We re-created this sub-panel and display both asparagine N+1 and N+2.
- Fig. 5A-B. Visual representation amended for clarity. Curves were compared by area under the curve, and statistical significance reported.
- In appendix fig. S6D, the Ponceau staining was mirrored compared to the Western Blot. We amended the new figure to have both with the same orientation.
- In appendix fig. S6A, curves are now compared using area under the curve.

Dr. Alessandro Esposito
Brunel University of London
Department of Biosciences
London
United Kingdom

Dear Dr. Esposito,

I am very pleased to accept your manuscript for publication in the next available issue of EMBO reports. Thank you for your contribution to our journal.

Yours sincerely,
